# Representation Balancing with Decomposed Patterns for Treatment Effect Estimation

## Abstract

Estimating treatment effects from observational data is subject to a problem of covariate shift caused by selection bias. Recent studies have attempted to mitigate this problem by group distance minimization, that is, balancing the distribution of representations between the treated and controlled groups. The rationale behind this is that learning balanced representations while preserving the predictive power of factual outcomes is expected to generalize to counterfactual inference. Inspired by this, we propose a new approach to better capture the patterns that contribute to representation balancing and outcome prediction. Specifically, we derive a theoretical bound that naturally ties the notion of propensity confusion to representation balancing, and further transform the balancing Patterns into Decompositions of Individual propensity confusion and Group distance minimization (PDIG). Moreover, we propose to decompose proxy features into Patterns of Pre-balancing and Balancing Representations (PPBR), as it is insufficient if only balanced representations are considered in outcome prediction. Extensive experiments on simulation and benchmark data confirm not only PDIG leads to mutual reinforcement between individual propensity confusion and group distance minimization, but also PPBR brings improvement to outcome prediction, especially to counterfactual inference. We believe these findings are heuristics for further investigation of what affects the generalizability of representation balancing models in counterfactual estimation.

## 1 Introduction

In the context of the ubiquity of personalized decision-making, causal inference has sparked a surge of research exploring causal machine learning in many disciplines, including economics and statistics (Wager & Athey, 2018; Athey & Wager, 2019; Farrell, 2015; Chernozhukov et al., 2018; Huang et al., 2021), healthcare (Qian et al., 2021; Bica et al., 2021a;b), and commercial applications (Guo et al., 2020b;c; Chu et al., 2021). The main problem of causal inference is the *treatment effect estimation*, which is tied to a fundamental hypothetical question: What would be the outcome if one received an alternative treatment? Answering this question requires the knowledge of *counterfactual outcomes*, but they can only be inferred from observational data, not directly obtained.

*Selection bias* presents a major challenge for estimating counterfactual outcomes (Guo et al., 2020a; Zhang et al., 2020; Yao et al., 2021). This problem is caused by the non-random treatment assignment, that is, treatment (e.g., vaccination) is usually determined by covariates (e.g., age) that also affect the outcome (e.g., infection rate) (Huang et al., 2022b). The probability of a person receiving treatment is well known as the *propensity score*, and the difference between each person's propensity score can inherently lead to a covariate shift problem, i.e., the distribution of covariates in the treated units is substantially different from that in the controlled ones. The covariate shift issue makes it more difficult to infer counterfactual outcomes from observational data (Yao et al., 2018; Hassanpour & Greiner, 2019a).

Recently, a line of representation balancing works has sought to alleviate the covariate shift problem by balancing the distribution between the treated group and the controlled group in the representation space (Shalit et al., 2017; Johansson et al., 2022). The rational insight behind these works is that the counterfactual estimation should rest on the accuracy of factual estimation while enforcing minimization of distributional discrepancy measured by the Integral Probability Metric (IPM) between

the treated and controlled units. However, there are two issues that remain to be resolved. First, Wasserstein distance (Cuturi & Doucet, 2014) is the most widely-adopted metric for group distance minimization (Shalit et al., 2017; Huang et al., 2022a; Zhou et al., 2022), whereas $\mathcal{H}$-divergence has still received little attention in causal representation learning though it is an important distance metric in other fields (Ben-David et al., 2006; 2010). Second, enforcing models to learn outcome predictors with only balanced representations may inadvertently weaken the predictive power of the outcome function (Zhang et al., 2020; Assaad et al., 2021; Huang et al., 2022a). We provide intuitive examples to illustrate the second issue in Section A.4.

The aforementioned issues motivate us to explore approaches to (i) improving factual outcome prediction without affecting learning balancing patterns or (ii) learning more effective balancing patterns without affecting factual outcome prediction. In this paper, we propose a new method, DIGNet, with learning decomposed patterns to achieve these two goals. The **contributions** are threefold: (1) We interpret representation balancing as a concept of propensity confusion and derive corresponding theoretical results based on $\mathcal{H}$-divergence to ensure its rationality; (2) DIGNet transforms the balancing Patterns into Decompositions of Individual propensity confusion and Group distance minimization (*PDIG*) to capture patterns beneficial to representation balancing, and we empirically find that the PDIG structure enables individual propensity confusion and group distance minimization to reinforce each other without affecting factual outcome prediction; (3) DIGNet decomposes representative features into Patterns of Pre-balancing and Balancing Representations (*PPBR*) to preserve patterns contributing to outcome modeling, and we experimentally confirm that the PPBR approach brings improvement to outcome prediction without affecting learning balancing patterns.

## 1.1 RELATED WORK

The presence of a covariate shift problem stimulates the line of representation balancing works (Johansson et al., 2016; Shalit et al., 2017; Johansson et al., 2022). These works aim to balance the distributions of representations between treated and controlled groups and simultaneously try to maintain representations predictive of factual outcomes. This idea is closely connected with domain adaptation. In particular, the individual treatment effect (ITE) error bound based on Wasserstein distance is similar to the generalization bound in Ben-David et al. (2010); Long et al. (2014); Shen et al. (2018). In addition to Wasserstein distance-based model, this paper derives a new ITE error bound based on $\mathcal{H}$-divergence (Ben-David et al., 2006; 2010; Ganin et al., 2016). Note that our theoretical results (Section 3.2) and experimental implementations (Section 4.1) differ greatly from Shalit et al. (2017) due to distinct definitions between Wasserstein distance and $\mathcal{H}$-divergence.

Another recent line of work investigates efficient neural network structures for treatment effect estimation. Kuang et al. (2017); Hassanpour & Greiner (2019b) extract the original covariates into treatment-specific factors, outcome-specific factors, and confounding factors; X-learner (Künzel et al., 2019) and R-learner (Nie & Wager, 2021) are developed beyond the classic S-learner and T-learner; Curth & van der Schaar (2021) leverage structures for end-to-end learners to counteract the inductive bias towards treatment effect estimation, as motivated by Makar et al. (2020).

The proposed DIGNet model is built on the PDIG structure and the PPBR approach. The PDIG structure is motivated by multi-task learning, where we design a framework incorporating two specific balancing patterns that share the same pre-balancing patterns. The PPBR approach is inspired by Zhang et al. (2020); Assaad et al. (2021); Huang et al. (2022a), where the authors argue that improperly balanced representations can be detrimental predictors for outcome modeling, since such representations can lose the original information that contributes to outcome prediction. Other representation learning methods relevant to treatment effect estimation include Louizos et al. (2017); Yao et al. (2018); Yoon et al. (2018); Shi et al. (2019); Du et al. (2021).

## 2 PRELIMINARIES

**Notations.** Suppose there are the $N$ i.i.d. random variable samples $\mathcal{D} = \left\{(\mathbf{X}_i, T_i, Y_i)\right\}_{i=1}^{N}$ with observed realizations $\{(\mathbf{x}_i, t_i, y_i)\}_{i=1}^{N}$, where there are $N_1$ treated units and $N_0$ controlled units. For each unit $i$, $\mathbf{X}_i \in \mathcal{X} \subset \mathbb{R}^d$ denotes $d$-dimensional covariates and $T_i \in \{0, 1\}$ denotes the binary treatment, with $e(\mathbf{x}_i) := p(T_i = 1 \mid \mathbf{X}_i = \mathbf{x}_i)$ defined as the propensity score (Rosenbaum & Rubin, 1983). Potential outcome framework (Rubin, 2005) defines the potential outcomes $Y^1, Y^0 \in \mathcal{Y} \subset \mathbb{R}$

for treatment $T = 1$ and $T = 0$, respectively. We let the observed outcome (factual outcome) be $Y = T \cdot Y^1 + (1-T) \cdot Y^0$. For $t \in \{0,1\}$, let $\tau^t(\mathbf{x}) := \mathbb{E}\left[Y^t \mid \mathbf{X} = \mathbf{x}\right]$ be a function of $Y^t$ w.r.t. $\mathbf{X}$, then our goal is to estimate the individual treatment effect (ITE) $\tau(\mathbf{x}) := \mathbb{E}\left[Y^1 - Y^0 \mid \mathbf{X} = \mathbf{x}\right] = \tau^1(\mathbf{x}) - \tau^0(\mathbf{x})$, and the average treatment effect (ATE) $\tau_{ATE} := \mathbb{E}\left[Y^1 - Y^0\right] = \int_{\mathcal{X}} \tau(\mathbf{x}) p(\mathbf{x}) d\mathbf{x}$.

## 2.1 PROBLEM SETUP

In causal representation balancing works, we denote representation space by $\mathcal{R} \subset \mathbb{R}^d$, and $\Phi : \mathcal{X} \to \mathcal{R}$ is assumed to be a twice-differentiable, one-to-one and invertible function with its inverse $\Psi : \mathcal{R} \to \mathcal{X}$ such that $\Psi(\Phi(\mathbf{x})) = \mathbf{x}$. The densities of the treated and controlled covariates are denoted by $p_{\mathbf{x}}^{T=1} = p^{T=1}(\mathbf{x}) := p(\mathbf{x} \mid T = 1)$ and $p_{\mathbf{x}}^{T=0} = p^{T=0}(\mathbf{x}) := p(\mathbf{x} \mid T = 0)$, respectively. Correspondingly, the densities of the treated and controlled covariates in the representation space are denoted by $p_{\Phi}^{T=1} = p_{\Phi}^{T=1}(\mathbf{r}) := p_{\Phi}(\mathbf{r} \mid T = 1)$ and $p_{\Phi}^{T=0} = p_{\Phi}^{T=0}(\mathbf{r}) := p_{\Phi}(\mathbf{r} \mid T = 0)$, respectively.

Our study is based on the potential outcome framework (Rubin, 2005). Assumption 1 states standard and necessary assumptions to ensure treatment effects are identifiable. Before proceeding with theoretical analysis, we also present the necessary terms and definitions in Definition 1.

**Assumption 1 (Consistency, Overlap, and Unconfoundedness)** *Consistency: If the treatment is $t$, then the observed outcome equals $Y^t$. Overlap: The propensity score is bounded away from 0 to 1: $0 < e(\mathbf{x}) < 1$. Unconfoundedness: $Y^t \perp\!\!\!\perp T \mid \mathbf{X}, \forall t \in \{0,1\}$.*

**Definition 1** *Let $h : \mathcal{R} \times \{0,1\} \to \mathcal{Y}$ be an hypothesis defined over the representation space $\mathcal{R}$ such that $h(\Phi(\mathbf{x}), t)$ estimates $y^t$, and $L : \mathcal{Y} \times \mathcal{Y} \to \mathbb{R}_+$ be a loss function (e.g., $L(y, y') = (y - y')^2$). If we define the expected loss for $(\mathbf{x}, t)$ as $\ell_{h,\Phi}(\mathbf{x}, t) = \int_{\mathcal{Y}} L(y^t, h(\Phi(\mathbf{x}), t)) p(y^t | \mathbf{x}) dy^t$, we then have factual and counterfactual losses, as well as them on the treated and controlled:*

$$\epsilon_F(h, \Phi) = \int_{\mathcal{X} \times \{0,1\}} \ell_{h,\Phi}(\mathbf{x}, t) p(\mathbf{x}, t) d\mathbf{x} dt, \quad \epsilon_{CF}(h, \Phi) = \int_{\mathcal{X} \times \{0,1\}} \ell_{h,\Phi}(\mathbf{x}, t) p(\mathbf{x}, 1-t) d\mathbf{x} dt,$$

$$\epsilon_F^{T=1}(h, \Phi) = \int_{\mathcal{X}} \ell_{h,\Phi}(\mathbf{x}, 1) p^{T=1}(\mathbf{x}) d\mathbf{x}, \qquad \epsilon_F^{T=0}(h, \Phi) = \int_{\mathcal{X}} \ell_{h,\Phi}(\mathbf{x}, 0) p^{T=0}(\mathbf{x}) d\mathbf{x},$$

$$\epsilon_{CF}^{T=1}(h, \Phi) = \int_{\mathcal{X}} \ell_{h,\Phi}(\mathbf{x}, 1) p^{T=0}(\mathbf{x}) d\mathbf{x}, \qquad \epsilon_{CF}^{T=0}(h, \Phi) = \int_{\mathcal{X}} \ell_{h,\Phi}(\mathbf{x}, 0) p^{T=1}(\mathbf{x}) d\mathbf{x}.$$

If we let $f(\mathbf{x}, t)$ be $h(\Phi(\mathbf{x}), t)$, where $f : \mathcal{X} \times \{0,1\} \to \mathcal{Y}$ is a prediction function for outcome, then the estimated ITE over $f$ is defined as $\hat{\tau}_f(\mathbf{x}) := f(\mathbf{x}, 1) - f(\mathbf{x}, 0)$. Finally, a better treatment effect estimation can be reformulated as a smaller error in Precision in the expected Estimation of Heterogeneous Effect (PEHE):

$$\epsilon_{PEHE}(f) = \int_{\mathcal{X}} L(\hat{\tau}_f(\mathbf{x}), \tau(\mathbf{x})) p(\mathbf{x}) d\mathbf{x}. \tag{1}$$

Here, $\epsilon_{PEHE}(f)$ can also be denoted by $\epsilon_{PEHE}(h, \Phi)$ if we let $f(\mathbf{x}, t)$ be $h(\Phi(\mathbf{x}), t)$.

## 3 THEORETICAL RESULTS

In this section, we first prove $\epsilon_{PEHE}$ is bounded by $\epsilon_F$ and $\epsilon_{CF}$ in Lemma 1. Next, we revisit the upper bound (Theorem 1) concerning the group distance minimization guided method in Section 3.1. Section 3.2 further discusses the theoretical results of the proposed individual propensity confusion guided method in Theorem 2. Proofs and additional theoretical results are deferred to Appendix.

**Lemma 1** *Let functions $h$ and $\Phi$ be as defined in Definition 1, and $L$ be the squared loss function. Recall that $\tau^t(\mathbf{x}) = \mathbb{E}\left[Y^t \mid \mathbf{X} = \mathbf{x}\right]$. Defining $\sigma_y^2 = \min\{\sigma_{y^t}^2(p(\mathbf{x}, t)), \sigma_{y^t}^2(p(\mathbf{x}, 1-t))\} \, \forall t \in \{0,1\}$, where $\sigma_{y^t}^2(p(\mathbf{x}, t)) = \int_{\mathcal{X} \times \{0,1\} \times \mathcal{Y}} (y^t - \tau^t(\mathbf{x}))^2 p(y^t | \mathbf{x}) p(\mathbf{x}, t) dy^t d\mathbf{x} dt$, we have*

$$\epsilon_{PEHE}(h, \Phi) \leq 2(\epsilon_{CF}(h, \Phi) + \epsilon_F(h, \Phi) - 2\sigma_y^2).$$

Note that similar results will hold as long as $L$ takes forms that satisfy the triangle inequality, but is not limited to the squared loss. For instance, we give the result for absolute loss in Lemma 6 in Appendix. This extends the result shown in Shalit et al. (2017) that $L$ only takes the squared loss.

### 3.1 GNet: Group Distance Minimization Guided Representation Balancing

Previous group distance minimization guided approaches seek representation balancing by minimizing the distance measured by the Integral Probability Metric (IPM) defined in Definition 2.

**Definition 2** *Let $\mathcal{G}$ be a function family consisting of functions $g : \mathcal{S} \to \mathbb{R}$. For a pair of distributions $p_1$, $p_2$ over $\mathcal{S}$, the Integral Probability Metric is defined as*

$$IPM_{\mathcal{G}}(p_1, p_2) := sup_{g \in \mathcal{G}} | \int_{\mathcal{S}} g(s)(p_1(s) - p_2(s))ds |.$$

If $\mathcal{G}$ is the family of 1-Lipschitz functions, we can obtain the so-called 1-Wasserstein distance, denoted by $Wass(p_1, p_2)$ (Sriperumbudur et al., 2012). Next, we give the bounds for counterfactual error $\epsilon_{CF}$ and ITE error $\epsilon_{PEHE}$ using Wasserstein distance in Theorem 1.

**Theorem 1** *Let $h$, $\Phi$, $\Psi$, $p_{\Phi}^{T=1}$, and $p_{\Phi}^{T=0}$ be as defined before. Let $L$ be the squared loss, $u := Pr(T = 1)$, and $\mathcal{G}$ be the family of 1-Lipschitz functions. Assuming that there exists a constant $B_{\Phi} \geq 0$ such that $g_{\Phi,h}(\mathbf{r}, t) := \frac{1}{B_{\Phi}} \cdot \ell_{h,\Phi}(\Psi(\mathbf{r}), t) \in \mathcal{G}$ for fixed $t \in \{0, 1\}$, we then have*

$$\epsilon_{CF}(h, \Phi) \leq (1 - u) \cdot \epsilon_F^{T=1}(h, \Phi) + u \cdot \epsilon_F^{T=0}(h, \Phi) + B_{\Phi} \cdot Wass(p_{\Phi}^{T=1}, p_{\Phi}^{T=0}), \quad (2)$$

$$\epsilon_{PEHE}(h, \Phi) \leq 2(\epsilon_F^{T=1}(h, \Phi) + \epsilon_F^{T=0}(h, \Phi) + B_{\Phi} \cdot Wass(p_{\Phi}^{T=1}, p_{\Phi}^{T=0}) - 2\sigma_y^2). \quad (3)$$

Theorem 1 will be identical to Shalit et al. (2017) if $L$ is the squared loss. Note, however, that similar results for Theorem 1 still hold as long as $L$ takes forms that satisfy the triangle inequality. For instance, we give the result for absolute loss in Theorem 1 in Appendix. We refer to the model as **GNet** (aka CFR-Wass in Shalit et al. (2017)) since it is based on group distance minimization.

### 3.2 INet: Individual Propensity Confusion Guided Representation Balancing

The propensity score is recognized central to treatment effect estimation because it characterizes the probability that one receives treatment (Rosenbaum & Rubin, 1983). Therefore, in addition to group distance minimization, the propensity score can be naturally used to identify if representations are adequately balanced, since representation balancing can be intuitively interpreted as propensity confusion, that is, when it is hard to distinguish whether each unit in the representation space is treated or controlled, the representations are thought adequately balanced. In section 4.1, we will demonstrate how minimizing $\mathcal{H}$-divergence (Definition 3) is empirically related to propensity confusion. Below, we first derive an ITE bound in Theorem 2 based on $\mathcal{H}$-divergence.

**Definition 3** *Given a pair of distributions $p_1$, $p_2$ over $\mathcal{S}$, and a hypothesis binary function class $\mathcal{H}$, the $\mathcal{H}$-divergence between $p_1$ and $p_2$ is defined as*

$$d_{\mathcal{H}}(p_1, p_2) := 2sup_{\eta \in \mathcal{H}} |Pr_{p_1}[\eta(s) = 1] - Pr_{p_2}[\eta(s) = 1]|. \quad (4)$$

**Theorem 2** *Let $h$, $\Phi$, $\Psi$, $p_{\Phi}^{T=1}$, and $p_{\Phi}^{T=0}$ be as defined before. Let $L$ be the squared loss, $u := Pr(T = 1)$, and $\mathcal{H}$ be the family of binary functions. Assuming that there exists a constant $K \geq 0$ such that $\int_{\mathcal{Y}} L(y, y')dy \leq K \; \forall y' \in \mathcal{Y}$, we then have*

$$\epsilon_{CF}(h, \Phi) \leq (1 - u) \cdot \epsilon_F^{T=1}(h, \Phi) + u \cdot \epsilon_F^{T=0}(h, \Phi) + \frac{K}{2} d_{\mathcal{H}}(p_{\Phi}^{T=1}, p_{\Phi}^{T=0}), \quad (5)$$

$$\epsilon_{PEHE}(h, \Phi) \leq \epsilon_F^{T=1}(h, \Phi) + \epsilon_F^{T=0}(h, \Phi) + \frac{K}{2} d_{\mathcal{H}}(p_{\Phi}^{T=1}, p_{\Phi}^{T=0}) - 2\sigma_y^2. \quad (6)$$

We can apply similar arguments in proving Theorem 2 to obtain the corresponding upper bounds as long as $L$ takes forms that satisfy the triangle inequality. For instance, the result for absolute loss and the proof of Theorem 2 are given in Theorem 2 in Appendix. We refer to the model as **INet** since it is based on individual propensity confusion.

## 4 Method

In this section, we will demonstrate how Theorem 2 is associated with propensity confusion, and further suggest a decomposition network DIGNet based on GNet and INet. Section 4.1 presents the objectives of standard representation balancing models, GNet and INet. Section 4.2 introduces PDIG and PPBR components for the representation balancing with decomposed patterns scheme, and gives the final objective of the proposed model DIGNet.

## 4.1 Representation Balancing without decomposed Patterns

In representation balancing models, given the input data tuples $(\mathbf{x}, \mathbf{t}, \mathbf{y}) = \{(\mathbf{x}_i, t_i, y_i)\}_{i=1}^N$, the original covariates $\mathbf{x}$ are extracted by some representation function $\Phi(\cdot)$, and representations $\Phi(\mathbf{x})$ are then fed into the outcome functions $h^1(\cdot) := h(\cdot, 1)$ and $h^0(\cdot) := h(\cdot, 0)$ that estimate the potential outcome $y^1$ and $y^0$, respectively. Finally, the factual outcome can be predicted by $h^t(\cdot) = th^1(\cdot) + (1-t)h^0(\cdot)$, and the corresponding outcome loss is

$$\mathcal{L}_y(\mathbf{x}, \mathbf{t}, \mathbf{y}; \Phi, h^t) = \frac{1}{N} \sum_{i=1}^N L(h^t(\Phi(\mathbf{x}_i)), y_i). \tag{7}$$

If a model does not have decomposition modes like GNet and INet, both outcome prediction and representation balancing will rely on the extracted features $\Phi(\mathbf{x})$. Below we will introduce the objectives of GNet and INet.

**Objective of GNet.** GNet learns the balancing patterns over $\Phi$ by minimizing the group distance loss $\mathcal{L}_G(\mathbf{x}, \mathbf{t}; \Phi) = Wass\left(\{\Phi(\mathbf{x}_i)\}_{i:t_i=0}, \{\Phi(\mathbf{x}_i)\}_{i:t_i=1}\right)$. If the original covariates $\mathbf{x}$ are extracted by the feature extractor $\Phi_E(\cdot)$, then the final objective of GNet is

$$\min_{\Phi_E, h^t} \quad \mathcal{L}_y(\mathbf{x}, \mathbf{t}, \mathbf{y}; \Phi_E, h^t) + \alpha_1 \mathcal{L}_G(\mathbf{x}, \mathbf{t}; \Phi_E). \tag{8}$$

For the convenience of the reader, we illustrate the structure of GNet in Figure 1(a).

**Objective of INet.** Next, we detail how Theorem 2 is related to propensity confusion and give the objective of INet. Let $\mathbb{I}(a)$ be an indicator function that gives 1 if $a$ is true, and $\mathcal{H}$ be the family of binary functions as defined in Theorem 2. The representation balancing seeks to ahieve a smaller empirical $\mathcal{H}$-divergence $\hat{d}_{\mathcal{H}}(p_\Phi^{T=1}, p_\Phi^{T=0})$ such that

$$\hat{d}_{\mathcal{H}}(p_\Phi^{T=1}, p_\Phi^{T=0}) = 2\left(1 - \min_{\eta \in \mathcal{H}} \left[\frac{1}{N_0} \sum_{i:t_i=0}^{N_0} \mathbb{I}[\eta(\Phi(\mathbf{x}_i)) = 0] + \frac{1}{N_1} \sum_{i:t_i=1}^{N_1} \mathbb{I}[\eta(\Phi(\mathbf{x}_i)) = 1]\right]\right). \tag{9}$$

The "min" part in equation 9 indicates that the optimal classifier $\eta^* \in \mathcal{H}$ minimizes the classification error between the estimated treatment $\eta^*(\Phi(\mathbf{x}_i))$ and the observed treatment $t_i$, i.e., discriminating whether $\Phi(\mathbf{x}_i)$ is controlled ($T = 0$) or treated ($T = 1$). As a result, $\hat{d}_{\mathcal{H}}(p_\Phi^{T=1}, p_\Phi^{T=0})$ will be large if $\eta^*$ can easily distinguish whether $\Phi(\mathbf{x}_i)$ is treated or controlled, i.e., the optimal classification error is small. In contrast, $\hat{d}_{\mathcal{H}}(p_\Phi^{T=1}, p_\Phi^{T=0})$ will be small if it is hard for $\eta^*$ to determine whether $\Phi(\mathbf{x}_i)$ is treated or controlled, i.e., the optimal classification error is large. Therefore, the prerequisite of a small $\mathcal{H}$-divergence is to find a map $\Phi$ such that any classifier $\eta \in \mathcal{H}$ will get confused about the probability of $\Phi(\mathbf{x}_i)$ being treated or controlled. To achieve this goal, we first define a discriminator $\pi(\mathbf{r}) : \mathcal{R} \to [0, 1]$ that estimates the propensity score of $\mathbf{r}$. The classification error for the $i^{th}$ individual can be empirically approximated by the cross-entropy loss between $\pi(\Phi(\mathbf{x}_i))$ and $t_i$:

$$\mathcal{L}_t(t_i, \pi(\Phi(\mathbf{x}_i))) = -\left[t_i \log \pi(\Phi(\mathbf{x}_i)) + (1 - t_i) \log(1 - \pi(\Phi(\mathbf{x}_i)))\right]. \tag{10}$$

To minimize the classification error in equation 9, we aim to find an optimal discriminator $\pi^*$ such that $\pi^*$ maximizes the probability that treatment is correctly classified of the total population:

$$\max_{\pi \in \mathcal{H}} \mathcal{L}_I(\mathbf{x}, \mathbf{t}; \Phi, \pi) = \max_{\pi \in \mathcal{H}} \left[-\frac{1}{N_0} \sum_{i:t_i=0}^{N_0} \mathcal{L}_t(t_i, \pi(\Phi(\mathbf{x}_i))) - \frac{1}{N_1} \sum_{i:t_i=1}^{N_1} \mathcal{L}_t(t_i, \pi(\Phi(\mathbf{x}_i)))\right]. \tag{11}$$

Given the feature extractor $\Phi_E(\cdot)$, the objective of INet can be formulated as a min-max game:

$$\min_{\Phi_E, h^t} \max_{\pi} \quad \mathcal{L}_y(\mathbf{x}, \mathbf{t}, \mathbf{y}; \Phi_E, h^t) + \alpha_2 \mathcal{L}_I(\mathbf{x}, \mathbf{t}; \Phi_E, \pi). \tag{12}$$

As stated in equation 12, INet achieves the representation balancing through a min-max formulation. In the maximization, the discriminator $\pi$ is trained to maximize the probability that treatment is correctly classified. This forces $\pi(\Phi_E(\mathbf{x}_i))$ closer to the true propensity score $e(\mathbf{x}_i)$. In the minimization, the feature extractor $\Phi_E$ is trained to fool the discriminator $\pi$. This confuses $\pi$ such

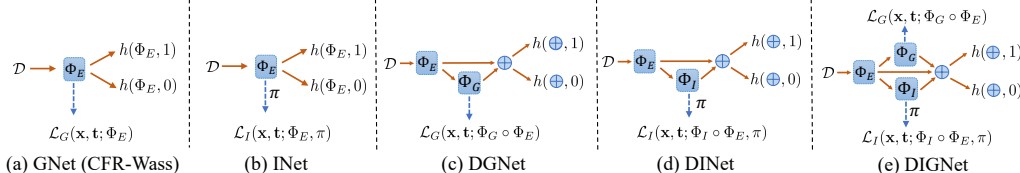

Figure 1: Illustrations of the network architecture of the five models studied in Section 5.

that $\pi(\Phi_E(\mathbf{x}_i))$ cannot correctly specify the true propensity score $e(\mathbf{x}_i)$. Eventually, the representations are balanced as it is difficult for $\pi$ to determine the propensity of $\Phi(\mathbf{x}_i)$ being treated or controlled. For the convenience of the reader, we illustrate the structure of INet in Figure 1(b). Note that though INet follows the strategy of approximating $\mathcal{H}$-divergence in Ganin et al. (2016), the theoretical derivations are completely different due to the significant differences (e.g., definitions, problem settings, and theoretical bounds.) between causal inference and domain adaptation.

## 4.2 Representation Balancing with Decomposed Patterns

**PDIG.** Previous demonstrations have shown that GNet is thriving and widely adopted, while INet is meaningful and interpretable. Nevertheless, there is no consensus on the absolute best approach, as each method has its strengths and weaknesses. To this end, we expect to capture more effective balancing patterns by turning the balancing Patterns into Decompositions of Individual propensity confusion and Group distance minimization (PDIG). More specifically, the covariates $\mathbf{x}$ are extracted by the feature extractor $\Phi_E(\cdot)$, and then $\Phi_E(\mathbf{x})$ are fed into the balancing networks $\Phi_G(\cdot)$ and $\Phi_I(\cdot)$ for group distance minimization and individual propensity confusion, respectively. Finally, the losses for the two separate balancing patterns are

$$\min_{\Phi_G} \mathcal{L}_G(\mathbf{x}, \mathbf{t}; \Phi_G \circ \Phi_E), \qquad \min_{\Phi_I} \max_{\pi} \mathcal{L}_I(\mathbf{x}, \mathbf{t}; \Phi_I \circ \Phi_E, \pi). \tag{13}$$

Here, $\circ$ denotes the composition of two functions, indicating that $\Phi(\cdot)$ in $\mathcal{L}_G(\mathbf{x}, \mathbf{t}; \Phi)$ and $\mathcal{L}_I(\mathbf{x}, \mathbf{t}; \Phi, \pi)$ are replaced by $\Phi_G(\Phi_E(\cdot))$ and $\Phi_I(\Phi_E(\cdot))$, respectively.

**PPBR.** Motivated by the discussion in Section 1, we aim to design a framework that is capable of capturing Patterns of Pre-balancing and Balancing Representations (PPBR). First, the representation balancing patterns $\Phi_G(\Phi_E(\mathbf{x}))$ and $\Phi_I(\Phi_E(\mathbf{x}))$ are learned over $\Phi_G$ and $\Phi_I$, while $\Phi_E$ is remained fixed as pre-balancing patterns. Furthermore, we concatenate the balancing representations $\Phi_G(\Phi_E(\mathbf{x}))$ and $\Phi_I(\Phi_E(\mathbf{x}))$ with the pre-balancing representations $\Phi_E(\mathbf{x})$ as attributes for outcome prediction. As a result, the proxy features used for outcome predictions are $\Phi_E(\mathbf{x}) \oplus \Phi_G(\Phi_E(\mathbf{x})) \oplus \Phi_I(\Phi_E(\mathbf{x}))$, where $\oplus$ indicates the concatenation by column. For example, if $\mathbf{a} = [1, 2]$ and $\mathbf{b} = [3, 4]$, then $\mathbf{a} \oplus \mathbf{b} = [1, 2, 3, 4]$.

Although we mainly investigate whether PDIG and PPBR are beneficial to treatment effect estimation in this paper, it would also be an interesting direction for future research to find out whether there exists any interaction or mutual reinforcement between them.

**Objective of DIGNet.** Combining with PDIG and PPBR, we develop a new model architecture, DIGNet, as illustrated in Figure 1(e). The objective of DIGNet is separated into four stages:

$$\min_{\Phi_G} \alpha_1 \mathcal{L}_G(\mathbf{x}, \mathbf{t}; \Phi_G \circ \Phi_E), \tag{14}$$

$$\max_{\pi} \alpha_2 \mathcal{L}_I(\mathbf{x}, \mathbf{t}; \Phi_I \circ \Phi_E, \pi), \tag{15}$$

$$\min_{\Phi_I} \alpha_2 \mathcal{L}_I(\mathbf{x}, \mathbf{t}; \Phi_I \circ \Phi_E, \pi), \tag{16}$$

$$\min_{\Phi_E, \Phi_I, \Phi_G, h^t} \mathcal{L}_y(\mathbf{x}, \mathbf{t}, \mathbf{y}; \Phi_E \oplus (\Phi_I \circ \Phi_E) \oplus (\Phi_G \circ \Phi_E), h^t). \tag{17}$$

Within each iteration, DIGNet manages to minimize the group distance via equation 14, and plays an adversarial game to achieve propensity confusion through equation 15 and equation 16. In equation 17, DIGNet updates both the pre-balancing and balancing patterns $\Phi_E, \Phi_I, \Phi_G$ along with the outcome function $h^t$ to minimize the outcome prediction loss.

**DGNet and DINet.** For further ablation studies, we also propose two models, DGNet and DINet. The two models can be considered as either DIGNet without PDIG, or GNet and INet with PPBR. The structures of DGNet and DINet are shown in Figure 1(c) and Figure 1(d), and the objectives of DGNet and DINet are deferred to Section A.6 in Appendix.

## 5 EXPERIMENTS

In non-randomized observational data, the ground truth of treatment effects is inaccessible due to the lack of counterfactuals. Therefore, we use simulated data and semi-synthetic benchmark data to test the performance of our methods and other baseline models. In this section, we mainly investigate two questions: **Q1.** Compared to DGNet and DINet without the PDIG structure, can DIGNet with the PDIG structure achieve a better representation balancing task? **Q2.** Are DGNet and DINet that involve PPBR capable of improving the performance on outcome prediction compared with standard representation balancing models such as GNet and INet?

### 5.1 EXPERIMENTAL SETTINGS

**Simulation data.** Previous causal inference works assess the model effectiveness by varying the distribution imbalance of covariates in treated and controlled groups at different levels (Yao et al., 2018; Yoon et al., 2018; Du et al., 2021). As suggested in Assaad et al. (2021), we draw 1000 observational data points from the following data generating strategy:

$$\mathbf{X}_i \sim \mathcal{N}(\mathbf{0}, \sigma^2 \cdot [\rho \mathbf{1}_p \mathbf{1}_p^{'}(1-\rho)\mathbf{I}_p]), \quad T_i \mid \mathbf{X}_i \sim \text{Bernoulli}(1/(1 + \exp(-\gamma \mathbf{X}_i))),$$

$$Y_i^0 = \boldsymbol{\beta}_{\mathbf{0}}' \mathbf{X}_i + \xi_i, \qquad Y_i^1 = \boldsymbol{\beta}_{\mathbf{1}}' \mathbf{X}_i + \xi_i, \qquad \xi_i \sim \mathcal{N}(0, 1).$$

Here, $\mathbf{1}_p$ denotes the $p$-dimensional all-ones vector and $\mathbf{I}_p$ denotes the identity matrix of size $p$. We fix $p = 10, \rho = 0.3, \sigma^2 = 2, \boldsymbol{\beta}_{\mathbf{0}}' = [0.3, ..., 0.3], \boldsymbol{\beta}_{\mathbf{1}}' = [1.3, ..., 1.3]$ and vary $\gamma \in \{0.25, 0.5, 0.75, 1, 1.5, 2, 3\}$ to yield different level of selection bias. As seen in Figure 6 in Appendix, selection bias becomes more severe with $\gamma$ increasing. For each $\gamma$, we repeat the above data generating process to generate 30 different datasets, with each dataset split by the ratio of $56\%/24\%/20\%$ as training/validation/test sets.

**Semi-synthetic data** The IHDP dataset is introduced by Hill (2011). This dataset consists of 747 samples with 25-dimensional covariates collected from real-world randomized experiments. Selection bias is created by removing some of treated samples. The goal is to estimate the effect of special visits (treatment) on cognitive scores (outcome). The potential outcomes are generated using the NPCI package Dorie (2021). We use the same 1000 datasets as used in Shalit et al. (2017), with each dataset split by the ratio of $63\%/27\%/10\%$ as training/validation/test sets.

**Models and metrics.** In simulation experiments, we perform comprehensive comparisons between INet, GNet, DINet, DGNet, and DIGNet in terms of the mean and standard error for the following metrics: $\sqrt{\epsilon_{PEHE}}$, $\sqrt{\epsilon_{CF}}$, and $\sqrt{\epsilon_F}$ with $L$ defined in Definition 1 being the squared loss, as well as the empirical approximations of $Wass(p_\Phi^{T=1}, p_\Phi^{T=0})$ and $d_{\mathcal{H}}(p_\Phi^{T=1}, p_\Phi^{T=0})$ (denoted by $Wass$ and $\hat{d}_{\mathcal{H}}$, respectively). Note that as shown in Figure 1, $Wass$ is over $\Phi_E$ for GNet while over $\Phi_G$ for DGNet and DIGNet; $\hat{d}_{\mathcal{H}}$ is over $\Phi_E$ for INet while over $\Phi_I$ for DINet and DIGNet. To analyze the source of gain, we fairly compare models by ensuring that each model shares the same hyperparameters, e.g., learning rate, the number of layers and units for $(\Phi_E, \Phi_G, \Phi_I, f^t)$, and $(\alpha_1, \alpha_2)$. Note that we apply an early stopping rule to all models as Shalit et al. (2017) do. In IHDP experiment, we use $\sqrt{\epsilon_{PEHE}}$, as well as an additional metric $\epsilon_{ATE} = |\hat{\tau}_{ATE} - \tau_{ATE}|$ to evaluate performances of various causal models (see them in Table 3). More descriptions of the implementation are detailed in Section A.5 of Appendix.

### 5.2 RESULTS AND ANALYSIS

**Varying selection bias.** We first make a general comparison between the models on datasets when the degree of covariate imbalance increases, and the relevant results are shown in Figure 2. There are three main observations: (1) DIGNet attains the lowest $\sqrt{\epsilon_{PEHE}}$ across all datasets, while GNet

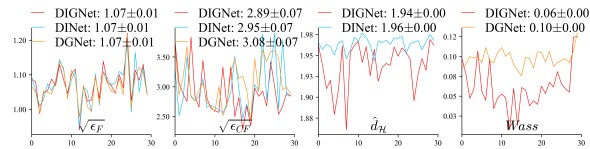

Figure 2: Plots of model performances on test set for different metrics as $\gamma$ varies in $\{0.25, 0.5, 0.75, 1, 1.5, 2, 3\}$. Each graph shows the average of 30 runs with standard errors shaded.

performs worse than other models; (2) DINet and DGNet outperform INet and GNet regarding $\sqrt{\epsilon_{CF}}$ and $\sqrt{\epsilon_{PEHE}}$; (3) INet, DINet, and DGNet perform similarly to DIGNet on factual outcome estimations ($\sqrt{\epsilon_F}$), but cannot compete with DIGNet in terms of counterfactual estimations ($\sqrt{\epsilon_{CF}}$); (4) DIGNet achieves smaller $\hat{d}_{\mathcal{H}}$ (or $Wass$) than DINet and INet (or DGNet and GNet), especially when the covariate shift problem is severe (e.g., when $\gamma > 1$).

In conclusion, the finding (2) reveals that *(i) the PPBR approach improves the predictive power of outcomes, especially for counterfactual outcomes*; and the findings (3) and (4) reveal that *(ii) the PDIG structure conduces to making group distance minimization and individual propensity confusion complementary and mutually reinforcing.*

**Source of gain.**    To further investigate the above findings, we choose the case with high selection bias ($\gamma = 3$) to explore the source of gain for PDIG and PPBR. We report model performances (mean $\pm$ std) averaged over 30 training and test sets in Table 1 and plot specific metrics of 30 runs on test set in Figure 3 and Figure 4. Below we discuss the source of gain in detail.

*(1) Ablation study for PDIG:* The PDIG structure is manifest to be effective in capturing balancing patterns. According to Figure 2, although DIGNet, DINet, and DGNet have comparable estimates of factual outcomes ($\sqrt{\epsilon_F}$), DIGNet can achieve more balanced representations regardless of the discrepancy is measured by $\hat{d}_{\mathcal{H}}$ or $Wass$. In particular, by comparing DIGNet with DGNet and DINet in Figure 3, we find that the PDIG structure does not affect the factual outcome estimation

Figure 3: Plots of model performances on test set for $\sqrt{\epsilon_F}$, $\sqrt{\epsilon_{CF}}$, $\hat{d}_{\mathcal{H}}$, and $Wass$ when $\gamma = 3$. Each graph plots the metric for 30 runs. Mean $\pm$ std of each metric averaged across 30 runs are reported on the top.

($\sqrt{\epsilon_F}$). Nevertheless, DIGNet achieves smaller $\hat{d}_{\mathcal{H}}$ with a $|1.94/1.96 - 1| = 1.0\%$ reduction (or $Wass$ with a $|0.06/0.10 - 1| = 40\%$ reduction) compared with DINet (or DGNet). This indicates that PDIG enables group distance minimization and individual propensity confusion to complement and reinforce each other, thereby learning better balancing patterns. This bring benefits with regard to counterfactual estimation. In particular, DIGNet reduces $\sqrt{\epsilon_{CF}}$ by $|2.89/2.95 - 1| = 2.0\%$ and $|2.89/3.08 - 1| = 6.2\%$ for DINet and DGNet, respectively. Thanks to the efficacy of PDIG in capturing balancing patterns, DIGNet shows superiority in treatment effect estimation ($\sqrt{\epsilon_{PEHE}}$ and $\epsilon_{ATE}$) compared to DGNet and DINet, as seen in Table 1.

*(2) Ablation study for PPBR:* The PPBR approach plays an essential role in outcome prediction, especially counterfactual inference. From Figure 4, we gain an important insight that the difference in learned representation balancing patterns, measured by $Wass$ (or $\hat{d}_{\mathcal{H}}$), between DGNet and GNet (or DINet and INet), is negligible. This means that PPBR has no impact on the representation balancing task. However, PPBR can improve the predictive power of factual outcomes, reducing $\sqrt{\epsilon_F}$ by $|1.07/1.12 - 1| = 4.5\%$ for GNet and $|1.07/1.08 - 1| = 0.9\%$ for INet. Such improvement is pronounced in counterfactual estimation, where $\sqrt{\epsilon_{CF}}$ is reduced by $|3.08/3.55 - 1| = 13.2\%$ for GNet and $|2.95/3.47 - 1| = 15.0\%$ for INet. Benefiting from the advantage of PPBR, the treatment effect errors ($\sqrt{\epsilon_{PEHE}}$ and $\epsilon_{ATE}$) attained by DINet and DGNet are significantly smaller than those attained by INet and GNet, as shown in Table 1.

Figure 4: Plots of model performances on test set for $\sqrt{\epsilon_F}$, $\sqrt{\epsilon_{CF}}$, $\hat{d}_{\mathcal{H}}$, and $Wass$ when $\gamma = 3$. Left graphs compare DGNet with GNet, and right graphs compare DINet with INet. Each graph plots the metric for 30 runs. Mean $\pm$ std of each metric averaged across 30 runs are reported on the top.

Table 1: Training- & test- set $\sqrt{\epsilon_{PEHE}}$ & $\epsilon_{ATE}$ when $\gamma = 3$. Mean $\pm$ standard error of 30 runs.

Table 2: Training- & test- set $\sqrt{\epsilon_{PEHE}}$ & $\epsilon_{ATE}$ on IHDP. Mean $\pm$ standard error of 100 runs.

|  | Training set | | Test set | |  | Training set | | Test set | |
|---|---|---|---|---|---|---|---|---|---|
|  | $\sqrt{\epsilon_{PEHE}}$ | $\epsilon_{ATE}$ | $\sqrt{\epsilon_{PEHE}}$ | $\epsilon_{ATE}$ |  | $\sqrt{\epsilon_{PEHE}}$ | $\epsilon_{ATE}$ | $\sqrt{\epsilon_{PEHE}}$ | $\epsilon_{ATE}$ |
| GNet | 3.30±0.15 | 2.58±0.14 | 3.30±0.16 | 2.59±0.14 | GNet | 0.71±0.15 | 0.12±0.01 | 0.77±0.18 | 0.15±0.02 |
| INet | 3.24±0.11 | 2.46±0.09 | 3.22±0.12 | 2.47±0.10 | INet | 0.66±0.09 | 0.13±0.01 | 0.72±0.11 | 0.15±0.02 |
| DGNet | 2.86±0.06 | 2.15±0.03 | 2.83±0.07 | 2.15±0.04 | DGNet | 0.53±0.07 | 0.11±0.01 | 0.60±0.09 | 0.13±0.01 |
| DINet | 2.70±0.06 | 2.12±0.04 | 2.69±0.08 | 2.13±0.05 | DINet | 0.57±0.12 | 0.13±0.01 | 0.60±0.11 | 0.14±0.01 |
| DIGNet | 2.66±0.07 | 2.04±0.05 | 2.63±0.07 | 2.03±0.04 | DIGNet | 0.42±0.02 | 0.11±0.01 | 0.45±0.04 | 0.12±0.01 |

**Comparisons on IHDP benchmark.** We first conduct an ablation study for PDIG and PPBR on 1-100 IHDP datasets and report the results in Table 2. Further, we undergo comparisons between DIGNet and other causal models on 1-1000 IHDP datasets and report the results in Table 3. Note that "-" indicates either the result is not reproducible or the original paper does not report relevant values. Table 2 shows that DINet and DGNet are superior to INet and GNet but inferior to DIGNet concerning treatment effect estimation, suggesting that each component of PDIG and PPBR is

Table 3: Training- & test- set $\sqrt{\epsilon_{PEHE}}$ & $\epsilon_{ATE}$ on IHDP. Mean $\pm$ standard error of 1000 runs.

|  | Training set | | Test set | |
|---|---|---|---|---|
|  | $\sqrt{\epsilon_{PEHE}}$ | $\epsilon_{ATE}$ | $\sqrt{\epsilon_{PEHE}}$ | $\epsilon_{ATE}$ |
| CEVAE (Louizos et al., 2017) | 2.7 ± .1 | .34 ± .01 | 2.6 ± .1 | .46 ± .02 |
| TARNet (Shalit et al., 2017) | .88 ± .0 | .26 ± .01 | .95 ± .0 | .28 ± .01 |
| SITE (Yao et al., 2018) | .69 ± .0 | .22 ± .01 | .75 ± .0 | .24 ± .01 |
| GANITE (Yoon et al., 2018) | 1.9 ± .4 | .43 ± .05 | 2.4 ± .4 | .49 ± .05 |
| Dragonnet (Shi et al., 2019) | 1.3 ± .4 | .14 ± .01 | 1.3 ± .5 | .20 ± .05 |
| BNN (Johansson et al., 2016) | 2.2 ± .1 | .37 ± .03 | 2.1 ± .1 | .42 ± .03 |
| CFR-Wass (GNet) (Shalit et al., 2017) | .73 ± .0 | .12 ± .01 | .81 ± .0 | .15 ± .01 |
| DKLITE (Zhang et al., 2020) | .52 ± .0 | – | .65 ± .03 | – |
| CFR-ISW (Hassanpour & Greiner, 2019a) | – | – | .70 ± .0 | .19 ± .03 |
| BWCFR-OW (Assaad et al., 2021) | – | – | .65 ± .0 | .18 ± .01 |
| BWCFR-MW (Assaad et al., 2021) | – | – | .63 ± .0 | .19 ± .01 |
| BWCFR-TruncIPW (Assaad et al., 2021) | – | – | .63 ± .0 | .19 ± .01 |
| MBRL (Huang et al., 2022a) | .52 ± .0 | .12 ± .01 | .57 ± .0 | .13 ± .01 |
| DIGNet (Ours) | **.42 ± .0** | **.11 ± .01** | **.45 ± .0** | **.12 ± .01** |

advantageous for treatment effect estimation. For example, on the test set, DINet reduces $\sqrt{\epsilon_{PEHE}}$ by $|0.60/0.72 - 1| = 16.7\%$ for INet, and DIGNet achieves $|0.45/0.60 - 1| = 25\%$ error reduction regarding $\sqrt{\epsilon_{PEHE}}$ for DINet. This is consistent with the findings before: PPBR and PDIG are beneficial to treatment effect estimation. Table 3 demonstrates that models that involve either propensity score or representation balancing (e.g., DKLITE, CFR-X, BWCFR-X, and MBRL) attain $\sqrt{\epsilon_{PEHE}}$ and $\epsilon_{ATE}$ of $0.57 \sim 0.70$ and $0.13 \sim 0.19$, respectively. Compared to the second-best method, DIGNet improves performance by $|0.45/0.57 - 1| = 21\%$ and $|0.12/0.13 - 1| = 7.7\%$ regarding $\sqrt{\epsilon_{PEHE}}$ and $\epsilon_{ATE}$, respectively, revealing the prominent outperformance of the proposed method. Moreover, it is noticeable that DIGNet achieves the lowest errors overwhelmingly across datasets and metrics, indicating that the proposed method has the most robust performance.

# 6 CONCLUSION

In this paper, we derive a theoretical ITE bound based on $\mathcal{H}$-divergence and connect representation balancing with the concept of propensity confusion. Furthermore, we propose the components of PDIG and PPBR, on which we construct a decomposition network structure DIGNet for treatment effect estimation. Comprehensive experiments verify that PDIG and PPBR follow different pathways to achieve the same goal of improving treatment effect estimation. In particular, PDIG helps the model better capture representation balancing patterns without affecting outcome prediction, while PPBR preserves patterns predictive of outcomes to enhance the outcome prediction without affecting balancing patterns. We believe that our findings constitute an important step towards the generalizability of representation balancing models in counterfactual estimation.

Promising directions for future work include discouraging redundancy of shared information of balancing patterns in the PDIG structure, improving the efficacy of optimizing DIGNet's objective, and exploring whether there exists an interaction between PDIG and PPBR.

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

## A APPENDIX

### A.1 PRELIMINARIES

We start by making some assumptions about the distribution we concern, and give some necessary definitions and results. We agree with the strong ignorability assumption, which assume that there exists a joint distribution $p(\mathbf{X}, T, Y^0, Y^1)$ such that conditioning on covariate $\mathbf{X}$, the potential outcomes $Y^0$, $Y^1$ are independent of $T$, i.e., $(Y^0, Y^1) \perp\!\!\!\perp T \mid \mathbf{X}$, and the propensity score $e(\mathbf{x}) := p(T = 1 \mid \mathbf{X} = \mathbf{x})$ is bounded away from 0 to 1, i.e. $0 < e(\mathbf{x}) < 1$. Recall that we also assume consistency, i.e., if the treatment is $t$, the observed outcome equals $Y^t$. These assumptions are crucial conditions that make individual treatment effect identifiable (Imbens & Wooldridge, 2009).

**Definition 1** *The individual treatment effect (ITE) for unit* $\mathbf{x}$ *is:*

$$\tau(\mathbf{x}) = \mathbb{E}\left[Y^1 - Y^0 \mid \mathbf{X} = \mathbf{x}\right].$$

Let $\tau^t(\mathbf{x}) := \mathbb{E}\left[Y^t \mid \mathbf{X} = \mathbf{x}\right]$, we have $\tau(\mathbf{x}) = \tau^1(\mathbf{x}) - \tau^0(\mathbf{x})$. Let $f : \mathcal{X} \times \{0, 1\} \to \mathcal{Y}$ be a prediction function.

**Definition 2** *The individual treatment effect estimate can be defined as:*

$$\hat{\tau}_f(\mathbf{x}) := f(\mathbf{x}, 1) - f(\mathbf{x}, 0).$$

**Definition 3** *(Hill, 2011) Let $L : \mathcal{Y} \times \mathcal{Y} \to \mathbb{R}_+$ be a loss function. The expected Precision in Estimation of Heterogeneous Effect (PEHE) loss of $f$ is:*

$$\epsilon_{PEHE}(f) = \int_{\mathcal{X}} L(\hat{\tau}_f(\mathbf{x}) - \tau(\mathbf{x}))p(\mathbf{x})d\mathbf{x}.$$

**Definition 4** *The covariates' distributions in the treated and controlled groups can be denoted by $p^{T=1}(\mathbf{x}) := p(\mathbf{x} \mid T = 1)$ and $p^{T=0}(\mathbf{x}) := p(\mathbf{x} \mid T = 0)$, respectively.*

In our causal representation balancing approach, we assume that the representation function $\Phi : \mathcal{X} \to \mathcal{R}$ is a twice-differentiable, one-to-one function, where $\mathcal{R} \subset \mathbb{R}^d$ is the representation space. Then, we can denote $\Psi : \mathcal{R} \to \mathcal{X}$ by the inverse of $\Phi$ and the induced distribution of $\mathbf{r}$ by $p_\Phi$.

**Definition 5** *The covariates' distributions in the treated and controlled groups over $\mathcal{R}$ can be denoted by $p_\Phi^{T=1}(\mathbf{r}) := p_\Phi(\mathbf{r} \mid T = 1)$ and $p_\Phi^{T=0}(\mathbf{r}) := p_\Phi(\mathbf{r} \mid T = 0)$, respectively.*

Let $h : \mathcal{R} \times \{0, 1\} \to \mathcal{Y}$ be an hypothesis defined over the representation space $\mathcal{R}$, such that $f(\mathbf{x}, t) = h(\Phi(\mathbf{x}), t)$.

**Definition 6** *The expected loss for the unit and treatment pair $(\mathbf{x}, t)$ is :*

$$\ell_{h,\Phi}(\mathbf{x}, t) = \int_{\mathcal{Y}} L(y^t, h(\Phi(\mathbf{x}), t))p(y^t|\mathbf{x})dy^t.$$

**Definition 7** *The expected factual loss and counterfactual losses of $h$ and $\Phi$ are, respectively:*

$$\epsilon_F(h, \Phi) = \int_{\mathcal{X} \times \{0,1\}} \ell_{h,\Phi}(\mathbf{x}, t)p(\mathbf{x}, t)d\mathbf{x}dt,$$

$$\epsilon_{CF}(h, \Phi) = \int_{\mathcal{X} \times \{0,1\}} \ell_{h,\Phi}(\mathbf{x}, t)p(\mathbf{x}, 1 - t)d\mathbf{x}dt.$$

**Definition 8** *The expected treated and control losses are:*

$$\epsilon_F^{T=1}(h, \Phi) = \int_{\mathcal{X}} \ell_{h,\Phi}(\mathbf{x}, 1)p^{T=1}(\mathbf{x})d\mathbf{x},$$

$$\epsilon_F^{T=0}(h, \Phi) = \int_{\mathcal{X}} \ell_{h,\Phi}(\mathbf{x}, 0)p^{T=0}(\mathbf{x})d\mathbf{x},$$

$$\epsilon_{CF}^{T=1}(h, \Phi) = \int_{\mathcal{X}} \ell_{h,\Phi}(\mathbf{x}, 1)p^{T=0}(\mathbf{x})d\mathbf{x},$$

$$\epsilon_{CF}^{T=0}(h, \Phi) = \int_{\mathcal{X}} \ell_{h,\Phi}(\mathbf{x}, 0)p^{T=1}(\mathbf{x})d\mathbf{x}.$$

Let $u := Pr(T = 1)$ be the proportion of treated in the population. We then have the result:

**Lemma 1**

$$\epsilon_F(h, \Phi) = u \cdot \epsilon_F^{T=1}(h, \Phi) + (1 - u) \cdot \epsilon_F^{T=0}(h, \Phi),$$

$$\epsilon_{CF}(h, \Phi) = (1 - u) \cdot \epsilon_{CF}^{T=1}(h, \Phi) + u \cdot \epsilon_{CF}^{T=0}(h, \Phi).$$

Noting that $p(\mathbf{x}, t) = u \cdot p^{T=1}(\mathbf{x}) + (1 - u) \cdot p^{T=0}(\mathbf{x})$, the results can be easily obtained from the Definitions 7 and 8.

**Definition 9** *Let $\mathcal{G}$ be a function family consisting of functions $g : \mathcal{S} \to \mathbb{R}$. For a pair of distributions $p_1$, $p_2$ over $\mathcal{S}$, define the Integral Probability Metric:*

$$IPM_{\mathcal{G}}(p_1, p_2) = \sup_{g \in \mathcal{G}} |\int_{\mathcal{S}} g(s)(p_1(s) - p_2(s))ds|.$$

Let $\mathcal{G}$ be the family of 1-Lipschitz functions, we obtain the so-called 1-Wasserstein distance between distributions, which we denote $Wass(p_1, p_2)$ (Sriperumbudur et al., 2012).

**Definition 10** *Given a pair of distributions $p_1$, $p_2$ over $\mathcal{S}$, and a hypothesis binary function class $\mathcal{H}$, the $\mathcal{H}$-divergence between $p_1$ and $p_2$ is*

$$d_{\mathcal{H}}(p_1, p_2) = 2 sup_{\eta \in \mathcal{H}} |Pr_{p_1}[\eta(s) = 1] - Pr_{p_2}[\eta(s) = 1]|.$$

**Lemma 2** *Let $\mathcal{G}$ in Definition 9 be the family of binary functions, we obtain half of $\mathcal{H}$-divergence.*

**Proof** *Let $\mathbb{I}(\cdot)$ denotes an indicator function.*

$$d_{\mathcal{H}}(p_1, p_2)$$

$$= 2 \sup_{\eta \in \mathcal{H}} \left| \int_{\eta(s)=1} (p_1(s) - p_2(s)) ds \right|$$

$$= 2 \sup_{\eta \in \mathcal{H}} \left| \int_{\mathcal{S}} \mathbb{I}(\eta(s) = 1)(p_1(s) - p_2(s)) ds \right|$$

$$= 2 \sup_{\eta \in \mathcal{H}} \left| \int_{\mathcal{S}} \eta(s)(p_1(s) - p_2(s)) ds \right| \tag{18}$$

*The last equation is because an indicator function is also a binary function.*

$\square$

## A.2 Bounds for conterfactual error $\epsilon_{CF}$

We first derive the counterfactual error bounds when using Wasserstein distance. The following Lemma 3 and corresponding proof is identical to the Lemma 1 in (Shalit et al., 2017).

**Lemma 3** *Let $\Phi : \mathcal{X} \to \mathcal{R}$ be an invertible representation with $\Psi$ being its inverse. Let $p_{\Phi}^{T=1}(\mathbf{r})$, $p_{\Phi}^{T=0}(\mathbf{r})$ be as defined before. Let $h : \mathcal{R} \times \{0, 1\} \to \mathcal{Y}$, $u := Pr(T = 1)$ and $\mathcal{G}$ be the family of 1-Lipschitz functions. Assume there exists a constant $B_{\Phi} \geq 0$, such that for $t = 0, 1$, the function $g_{\Phi,h}(\mathbf{r}, t) := \frac{1}{B_{\Phi}} \cdot \ell_{h,\Phi}(\Psi(\mathbf{r}), t) \in \mathcal{G}$. Then we have:*

$$\epsilon_{CF}(h, \Phi) \leq (1 - u) \cdot \epsilon_F^{T=1}(h, \Phi) + u \cdot \epsilon_F^{T=0}(h, \Phi) + B_{\Phi} \cdot Wass(p_{\Phi}^{T=1}, p_{\Phi}^{T=0}).$$

**Proof**

$$\epsilon_{CF}(h, \Phi) - [(1 - u) \cdot \epsilon_F^{T=1}(h, \Phi) + u \cdot \epsilon_F^{T=0}(h, \Phi)]$$

$$= [(1 - u) \cdot \epsilon_{CF}^{T=1}(h, \Phi) + u \cdot \epsilon_{CF}^{T=0}(h, \Phi)] - [(1 - u) \cdot \epsilon_F^{T=1}(h, \Phi) + u \cdot \epsilon_F^{T=0}(h, \Phi)]$$

$$= (1 - u) \cdot [\epsilon_{CF}^{T=1}(h, \Phi) - \epsilon_F^{T=1}(h, \Phi)] + u \cdot [\epsilon_{CF}^{T=0}(h, \Phi) - \epsilon_F^{T=0}(h, \Phi)]$$

$$= (1 - u) \int_{\mathcal{X}} \ell_{h,\Phi}(\mathbf{x}, 1)(p^{T=0}(\mathbf{x}) - p^{T=1}(\mathbf{x})) d\mathbf{x} + u \int_{\mathcal{X}} \ell_{h,\Phi}(\mathbf{x}, 0)(p^{T=1}(\mathbf{x}) - p^{T=0}(\mathbf{x})) d\mathbf{x} \tag{19}$$

$$= (1 - u) \int_{\mathcal{R}} \ell_{h,\Phi}(\Psi(\mathbf{r}), 1)(p_{\Phi}^{T=0}(\mathbf{r}) - p_{\Phi}^{T=1}(\mathbf{r})) d\mathbf{r} + u \int_{\mathcal{R}} \ell_{h,\Phi}(\Psi(\mathbf{r}), 0)(p_{\Phi}^{T=1}(\mathbf{r}) - p_{\Phi}^{T=0}(\mathbf{r})) d\mathbf{r} \tag{20}$$

$$\leq B_{\Phi} \cdot (1 - u) \int_{\mathcal{R}} \frac{1}{B_{\Phi}} \ell_{h,\Phi}(\Psi(\mathbf{r}), 1)(p_{\Phi}^{T=0}(\mathbf{r}) - p_{\Phi}^{T=1}(\mathbf{r})) d\mathbf{r}$$

$$+ B_{\Phi} \cdot u \int_{\mathcal{R}} \frac{1}{B_{\Phi}} \ell_{h,\Phi}(\Psi(\mathbf{r}), 0)(p_{\Phi}^{T=1}(\mathbf{r}) - p_{\Phi}^{T=0}(\mathbf{r})) d\mathbf{r}$$

$$\leq B_{\Phi} \cdot (1 - u) \sup_{g \in \mathcal{G}} \left| \int_{\mathcal{R}} g(\mathbf{r})(p_{\Phi}^{T=0}(\mathbf{r}) - p_{\Phi}^{T=1}(\mathbf{r})) d\mathbf{r} \right|$$

$$+ B_{\Phi} \cdot u \cdot \sup_{g \in \mathcal{G}} \left| \int_{\mathcal{R}} g(\mathbf{r})(p_{\Phi}^{T=1}(\mathbf{r}) - p_{\Phi}^{T=0}(\mathbf{r})) d\mathbf{r} \right| \tag{21}$$

$$= B_{\Phi} \cdot Wass(p_{\Phi}^{T=1}, p_{\Phi}^{T=0}) \tag{22}$$

*Equation (19) is by Definition 8; equation (20) is by the change of formula, $p_{\Phi}^{T=0}(\mathbf{r}) = p^{T=0}(\Psi(\mathbf{r}))J_{\Psi}(\mathbf{r})$, $p_{\Phi}^{T=1}(\mathbf{r}) = p^{T=1}(\Psi(\mathbf{r}))J_{\Psi}(\mathbf{r})$, where $J_{\Psi}(\mathbf{r})$ is the absolute of the determinant of the Jacobian of $\Psi(\mathbf{r})$; inequality (21) is by the premise that $\frac{1}{B_{\Phi}} \cdot \ell_{h,\Phi}(\Psi(\mathbf{r}), t) \in \mathcal{G}$ for $t = 0, 1$, and (22) is by Definition 9 of an IPM.* □

The crucial condition in Lemma 3 is that $g_{\Phi,h}(\mathbf{r}, t) := \frac{1}{B_{\Phi}} \cdot \ell_{h,\Phi}(\Psi(\mathbf{r}), t) \in \mathcal{G}$. Bounds for $B_{\Phi}$ can be given to evaluate this constant when under more assumptions about the loss function $L$, the Lipschitz constants of $p(y^t|\mathbf{x})$, $h$, and the condition number of the Jacobian of $\Phi$. These assumptions and the specific bounds for $B_{\Phi}$ can be seen in supplement Section A.3 of (Shalit et al., 2017).

Now we turn to derive the counterfactual error bounds for the $\mathcal{H}$-divergence case.

**Assumption 1** *There exists a constant $K$ such that for all $y_2 \in \mathcal{Y}$, $\int_{\mathcal{Y}} L(y_1, y_2)dy_1 \leq K$.*

**Lemma 4** *Let $\Phi : \mathcal{X} \to \mathcal{R}$ be an invertible representation with $\Psi$ being its inverse. Let $p_{\Phi}^{T=1}(\mathbf{r})$, $p_{\Phi}^{T=0}(\mathbf{r})$ be as defined before. Let $h : \mathcal{R} \times \{0, 1\} \to \mathcal{Y}$, $u := Pr(T = 1)$ and $\mathcal{H}$ be the family of binary functions. Assume loss function $L$ obeys the Assumption 1. Then we have:*

$$\epsilon_{CF}(h, \Phi) \leq (1 - u) \cdot \epsilon_F^{T=1}(h, \Phi) + u \cdot \epsilon_F^{T=0}(h, \Phi) + \frac{K}{2} d_{\mathcal{H}}(p_{\Phi}^{T=1}, p_{\Phi}^{T=0}).$$

**Proof**

$$\epsilon_{CF}(h, \Phi) - [(1 - u) \cdot \epsilon_F^{T=1}(h, \Phi) + u \cdot \epsilon_F^{T=0}(h, \Phi)]$$

$$= (1 - u) \int_{\mathcal{R}} \ell_{h,\Phi}(\Psi(\mathbf{r}), 1)(p_{\Phi}^{T=0}(\mathbf{r}) - p_{\Phi}^{T=1}(\mathbf{r}))d\mathbf{r} + u \int_{\mathcal{R}} \ell_{h,\Phi}(\Psi(\mathbf{r}), 0)(p_{\Phi}^{T=1}(\mathbf{r}) - p_{\Phi}^{T=0}(\mathbf{r}))d\mathbf{r}$$

$$\tag{23}$$

$$\leq (1 - u) \int_{p_{\Phi}^{T=0} > p_{\Phi}^{T=1}} \ell_{h,\Phi}(\Psi(\mathbf{r}), 1)(p_{\Phi}^{T=0}(\mathbf{r}) - p_{\Phi}^{T=1}(\mathbf{r}))d\mathbf{r}$$

$$+ u \int_{p_{\Phi}^{T=1} > p_{\Phi}^{T=0}} \ell_{h,\Phi}(\Psi(\mathbf{r}), 0)(p_{\Phi}^{T=1}(\mathbf{r}) - p_{\Phi}^{T=0}(\mathbf{r}))d\mathbf{r} \tag{24}$$

$$\leq (1 - u)K \int_{p_{\Phi}^{T=0} > p_{\Phi}^{T=1}} (p_{\Phi}^{T=0}(\mathbf{r}) - p_{\Phi}^{T=1}(\mathbf{r}))d\mathbf{r} + u \cdot K \int_{p_{\Phi}^{T=1} > p_{\Phi}^{T=0}} (p_{\Phi}^{T=1}(\mathbf{r}) - p_{\Phi}^{T=0}(\mathbf{r}))d\mathbf{r}$$

$$\tag{25}$$

$$= (1 - u)K \int_{\mathcal{R}} \mathbb{I}(p_{\Phi}^{t=0} > p_{\Phi}^{T=1})(p_{\Phi}^{T=0}(\mathbf{r}) - p_{\Phi}^{T=1}(\mathbf{r}))d\mathbf{r}$$

$$+ u \cdot K \int_{\mathcal{R}} \mathbb{I}(p_{\Phi}^{T=1} > p_{\Phi}^{T=0})(p_{\Phi}^{T=1}(\mathbf{r}) - p_{\Phi}^{T=0}(\mathbf{r}))d\mathbf{r}$$

$$\leq (1 - u)K \sup_{\eta \in \mathcal{H}} | \int_{\mathcal{R}} \eta(\mathbf{r})(p_{\Phi}^{T=1}(\mathbf{r}) - p_{\Phi}^{T=0}(\mathbf{r}))d\mathbf{r}|$$

$$+ u \cdot K \cdot \sup_{\eta \in \mathcal{H}} | \int_{\mathcal{R}} \eta(\mathbf{r})(p_{\Phi}^{T=1}(\mathbf{r}) - p_{\Phi}^{T=0}(\mathbf{r}))d\mathbf{r}| \tag{26}$$

$$\leq K \cdot \sup_{\eta \in \mathcal{H}} | \int_{\mathcal{R}} \eta(\mathbf{r})((p_{\Phi}^{T=1}(\mathbf{r}) - p_{\Phi}^{T=0}(\mathbf{r})))d\mathbf{r}|$$

$$= \frac{K}{2} d_{\mathcal{H}}(p_{\Phi}^{T=1}, p_{\Phi}^{T=0}) \tag{27}$$

*Equation (23) is same to equation (20); equation (24) is by $\ell_{h,\Phi} \geq 0$ for all $\mathbf{r}$ and $t$; inequality (25) is by Definition 6 and Assumption 1; inequality (26) is because an indicator function is also a binary function; equation (27) is by (18) in Lemma 2.* □

### A.3 BOUNDS FOR THE PEHE LOSS $\epsilon_{PEHE}$

We first state two lemmas for $\epsilon_{PEHE}$ with respect to two different loss functions: the squared loss and the absolute loss. In fact, similar lemmas hold for loss functions that satisfy the (relaxed) triangle inequalities.

**Definition 11** *The expected variance of $y^t$ with regard to $p(\mathbf{x}, t)$ is:*

$$\sigma^2_{y^t}(p(\mathbf{x}, t)) = \int_{\mathcal{X} \times \{0,1\} \times \mathcal{Y}} (y^t - \tau^t(\mathbf{x}))^2 p(y^t|\mathbf{x}) p(\mathbf{x}, t) dy^t d\mathbf{x} dt,$$

*and define:*

$$\sigma^2_y = \min\{\sigma^2_{y^t}(p(\mathbf{x}, t)), \sigma^2_{y^t}(p(\mathbf{x}, 1 - t))\}.$$

**Lemma 5** *Let loss function $L$ be the squared loss, $L(y_1, y_2) = (y_1 - y_2)^2$. For any function $f : \mathcal{X} \times \{0,1\} \to \mathcal{Y}$, and distribution $p(\mathbf{x}, t)$ over $\mathcal{X} \times \{0,1\}$:*

$$\epsilon_{PEHE}(h, \Phi) \le 2(\epsilon_{CF}(h, \Phi) + \epsilon_F(h, \Phi) - 2\sigma^2_y)$$

**Proof** *We denote $\epsilon_{PEHE}(f) = \epsilon_{PEHE}(h, \Phi)$, $\epsilon_F(f) = \epsilon_F(h, \Phi)$, $\epsilon_{CF}(f) = \epsilon_{CF}(h, \Phi)$ for $f(\mathbf{x}, t) = h(\Phi(\mathbf{x}), t)$.*

$$\epsilon_{PEHE}(f)$$
$$= \int_{\mathcal{X}} ((f(\mathbf{x}, 1) - f(\mathbf{x}, 0)) - (\tau^1(\mathbf{x}) - \tau^0(\mathbf{x})))^2 p(\mathbf{x}) d\mathbf{x}$$
$$\le 2 \int_{\mathcal{X}} ((f(\mathbf{x}, 1) - \tau^1(\mathbf{x}))^2 + (f(\mathbf{x}, 0) - \tau^0(\mathbf{x}))^2) p(\mathbf{x}) d\mathbf{x} \tag{28}$$
$$= 2 \int_{\mathcal{X}} (f(\mathbf{x}, 1) - \tau^1(\mathbf{x}))^2 p(\mathbf{x}, T = 1) d\mathbf{x} + 2 \int_{\mathcal{X}} (f(\mathbf{x}, 0) - \tau^0(\mathbf{x}))^2 p(\mathbf{x}, T = 0) d\mathbf{x}$$
$$+ 2 \int_{\mathcal{X}} (f(\mathbf{x}, 1) - \tau^1(\mathbf{x}))^2 p(\mathbf{x}, T = 0) d\mathbf{x} + 2 \int_{\mathcal{X}} (f(\mathbf{x}, 0) - \tau^0(\mathbf{x}))^2 p(\mathbf{x}, T = 1) d\mathbf{x} \tag{29}$$
$$= 2 \int_{\mathcal{X} \times \{0,1\}} (f(\mathbf{x}, t) - \tau^t(\mathbf{x}))^2 p(\mathbf{x}, t) d\mathbf{x} dt + 2 \int_{\mathcal{X} \times \{0,1\}} (f(\mathbf{x}, t) - \tau^t(\mathbf{x}))^2 p(\mathbf{x}, 1 - t) d\mathbf{x} dt.$$

*Inequality (28) is because the relaxed triangle inequality, $(x + y)^2 \le 2(x^2 + y^2)$; equation (29) is because $p(\mathbf{x}) = p(\mathbf{x}, T = 0) + p(\mathbf{x}, T = 1)$.*

$$\epsilon_F(f)$$
$$= \int_{\mathcal{X} \times \{0,1\} \times \mathcal{Y}} (f(\mathbf{x}, t) - y^t)^2 p(y^t|\mathbf{x}) p(\mathbf{x}, t) dy^t d\mathbf{x} dt$$
$$= \int_{\mathcal{X} \times \{0,1\} \times \mathcal{Y}} (f(\mathbf{x}, t) - \tau^t(\mathbf{x}))^2 p(y^t|\mathbf{x}) p(\mathbf{x}, t) dy^t d\mathbf{x} dt$$
$$+ \int_{\mathcal{X} \times \{0,1\} \times \mathcal{Y}} (\tau^t(\mathbf{x}) - y^t)^2 p(y^t|\mathbf{x}) p(\mathbf{x}, t) dy^t d\mathbf{x} dt$$
$$+ 2 \int_{\mathcal{X} \times \{0,1\} \times \mathcal{Y}} (f(\mathbf{x}, t) - \tau^t(\mathbf{x}))(\tau^t(\mathbf{x}) - y^t) p(y^t|\mathbf{x}) p(\mathbf{x}, t) dy^t d\mathbf{x} dt \tag{30}$$
$$= \int_{\mathcal{X} \times \{0,1\}} (f(\mathbf{x}, t) - \tau^t(\mathbf{x}))^2 p(\mathbf{x}, t) d\mathbf{x} dt + \sigma^2_{y^t}(p(\mathbf{x}, t)) \tag{31}$$

*Equation (31) is by Definition 11 and last term in equation (30) equals to zero, since $\tau^t(\mathbf{x}) = \int_{\mathcal{Y}} y^t p(y^t|\mathbf{x}) dy_t$. A similar result can be obtained for $\epsilon_{CF}$:*

$$\epsilon_{CF}(f) = \int_{\mathcal{X} \times \{0,1\}} (f(\mathbf{x}, t) - \tau^t(\mathbf{x}))^2 p(\mathbf{x}, 1 - t) d\mathbf{x} dt + \sigma^2_{y^t}(p(\mathbf{x}, 1 - t)).$$

*Combining these results and Definition 11, we have*

$$\epsilon_{PEHE}(h, \Phi) \le 2(\epsilon_F(f) - \sigma^2_{y^t}(p(\mathbf{x}, t))) + 2(\epsilon_{CF}(f) - \sigma^2_{y^t}(p(\mathbf{x}, 1 - t)))$$
$$\le 2(\epsilon_{CF}(h, \Phi) + \epsilon_F(h, \Phi) - 2\sigma^2_y).$$

$\square$

For the absolute loss $L(y_1, y_2) = |y_1 - y_2|$ that satisfies triangle inequality, the upper bound in Lemma 5 will replace the standard deviation $\sigma_y^2$ by mean absolute deviation $A_y$.

**Definition 12** *The mean absolute deviation of $y^t$ with regard to $p(\mathbf{x}, t)$ is:*

$$A_{y^t}(p(\mathbf{x}, t)) = \int_{\mathcal{X} \times \{0,1\} \times \mathcal{Y}} |y^t - \tau^t(\mathbf{x})| p(y^t|\mathbf{x}) p(\mathbf{x}, t) dy^t d\mathbf{x} dt,$$

*and define:*

$$A_y = \min\{A_{y^t}(p(\mathbf{x}, t)), A_{y^t}(p(\mathbf{x}, 1 - t))\}.$$

**Lemma 6** *Let loss function $L$ be the absolute loss, $L(y_1, y_2) = |y_1 - y_2|$. For any function $f : \mathcal{X} \times \{0, 1\} \to \mathcal{Y}$, and distribution $p(\mathbf{x}, t)$ over $\mathcal{X} \times \{0, 1\}$:*

$$\epsilon_{PEHE}(h, \Phi) \leq \epsilon_{CF}(h, \Phi) + \epsilon_F(h, \Phi) - 2A_y.$$

**Proof** *Recall that $\epsilon_{PEHE}(f) = \epsilon_{PEHE}(h, \Phi)$, $\epsilon_F(f) = \epsilon_F(h, \Phi)$, $\epsilon_{CF}(f) = \epsilon_{CF}(h, \Phi)$ for $f(\mathbf{x}, t) = h(\Phi(\mathbf{x}), t)$.*

$$
\begin{aligned}
&\epsilon_{PEHE}(f) \\
&= \int_{\mathcal{X}} |(f(\mathbf{x}, 1) - f(\mathbf{x}, 0)) - (\tau^1(\mathbf{x}) - \tau^0(\mathbf{x}))| p(\mathbf{x}) d\mathbf{x} \\
&\leq \int_{\mathcal{X}} (|f(\mathbf{x}, 1) - \tau^1(\mathbf{x})| + |f(\mathbf{x}, 0) - \tau^0(\mathbf{x})|) p(\mathbf{x}) d\mathbf{x} \quad (32) \\
&= \int_{\mathcal{X}} |f(\mathbf{x}, 1) - \tau^1(\mathbf{x})| p(\mathbf{x}, T = 1) d\mathbf{x} + \int_{\mathcal{X}} |f(\mathbf{x}, 0) - \tau^0(\mathbf{x})| p(\mathbf{x}, T = 0) d\mathbf{x} \\
&\quad + \int_{\mathcal{X}} |f(\mathbf{x}, 1) - \tau^1(\mathbf{x})| p(\mathbf{x}, T = 0) d\mathbf{x} + \int_{\mathcal{X}} |f(\mathbf{x}, 0) - \tau^0(\mathbf{x})| p(\mathbf{x}, T = 1) d\mathbf{x} \quad (33) \\
&= \int_{\mathcal{X} \times \{0,1\}} |f(\mathbf{x}, t) - \tau^t(\mathbf{x})| p(\mathbf{x}, t) d\mathbf{x} dt + \int_{\mathcal{X} \times \{0,1\}} |f(\mathbf{x}, t) - \tau^t(\mathbf{x})| p(\mathbf{x}, 1 - t) d\mathbf{x} dt.
\end{aligned}
$$

*Inequality (32) is because triangle inequality, $|x + y| \leq |x| + |y|$; equation (33) is because $p(\mathbf{x}) = p(\mathbf{x}, T = 0) + p(\mathbf{x}, T = 1)$.*

$$
\begin{aligned}
&\epsilon_F(f) \\
&= \int_{\mathcal{X} \times \{0,1\} \times \mathcal{Y}} |f(\mathbf{x}, t) - y^t| p(y^t|\mathbf{x}) p(\mathbf{x}, t) dy^t d\mathbf{x} dt \\
&\leq \int_{\mathcal{X} \times \{0,1\} \times \mathcal{Y}} |f(\mathbf{x}, t) - \tau^t(\mathbf{x})| p(y^t|\mathbf{x}) p(\mathbf{x}, t) dy^t d\mathbf{x} dt \\
&\quad + \int_{\mathcal{X} \times \{0,1\} \times \mathcal{Y}} |\tau^t(\mathbf{x}) - y^t| p(y^t|\mathbf{x}) p(\mathbf{x}, t) dy^t d\mathbf{x} dt \quad (34) \\
&= \int_{\mathcal{X} \times \{0,1\}} |f(\mathbf{x}, t) - \tau^t(\mathbf{x})| p(\mathbf{x}, t) d\mathbf{x} dt + A_{y^t}(p(\mathbf{x}, t)). \quad (35)
\end{aligned}
$$

*Inequality (34) is also because $|x + y| \leq |x| + |y|$, equation (35) is by Definition 12. A similar result can be obtained for $\epsilon_{CF}$:*

$$\epsilon_{CF}(f) = \int_{\mathcal{X} \times \{0,1\}} |f(\mathbf{x}, t) - \tau^t(\mathbf{x})| p(\mathbf{x}, 1 - t) d\mathbf{x} dt + A_{y^t}(p(\mathbf{x}, 1 - t)).$$

*Combining these results and Definition 12, we have*

$$
\begin{aligned}
\epsilon_{PEHE}(h, \Phi) &\leq \epsilon_F(f) - A_{y^t}(p(\mathbf{x}, t)) + \epsilon_{CF}(f) - A_{y^t}(p(\mathbf{x}, 1 - t)) \\
&\leq \epsilon_{CF}(h, \Phi) + \epsilon_F(h, \Phi) - 2A_y.
\end{aligned}
$$

$\square$

We summarize the upper bounds of $\epsilon_{CF}$ and $\epsilon_{PEHE}$ above, and give the final bounds for these two distance using the squared and absolute loss, respectively.

**Theorem 1** *Let $\Phi : \mathcal{X} \to \mathcal{R}$ be an invertible representation with $\Psi$ being its inverse. Let $p_\Phi^{T=1}(\mathbf{r})$, $p_\Phi^{T=0}(\mathbf{r})$ be as defined before. Let $h : \mathcal{R} \times \{0,1\} \to \mathcal{Y}$, $u := Pr(T = 1)$ and $\mathcal{G}$ be the family of 1-Lipschitz functions. Assume there exists a constant $B_\Phi \geq 0$, such that for $t = 0, 1$, the function $g_{\Phi,h}(\mathbf{r}, t) := \frac{1}{B_\Phi} \cdot \ell_{h,\Phi}(\Psi(\mathbf{r}), t) \in \mathcal{G}$.*

*Let loss function $L$ be the squared loss, $L(y_1, y_2) = (y_1 - y_2)^2$. Then we have:*

$$\epsilon_{PEHE}(h, \Phi)$$
$$\leq 2(\epsilon_{CF}(h, \Phi) + \epsilon_F(h, \Phi) - 2\sigma_y^2) \tag{36}$$
$$\leq 2(\epsilon_F^{T=1}(h, \Phi) + \epsilon_F^{T=0}(h, \Phi) + B_\Phi \cdot Wass(p_\Phi^{T=1}, p_\Phi^{T=0}) - 2\sigma_y^2) \tag{37}$$

*Let loss function $L$ be the absolute loss, $L(y_1, y_2) = |y_1 - y_2|$. Then we have:*

$$\epsilon_{PEHE}(h, \Phi)$$
$$\leq \epsilon_{CF}(h, \Phi) + \epsilon_F(h, \Phi) - 2A_y \tag{38}$$
$$\leq \epsilon_F^{T=1}(h, \Phi) + \epsilon_F^{T=0}(h, \Phi) + B_\Phi \cdot Wass(p_\Phi^{T=1}, p_\Phi^{T=0}) - 2A_y \tag{39}$$

**Proof** *Inequality (36) is by Lemma 5, inequality (37) is by Lemma 1 and Lemma 3; Inequality (38) is by Lemma 6, inequality (39) is by Lemma 1 and Lemma 3;* $\qquad\square$

**Theorem 2** *Let $\Phi : \mathcal{X} \to \mathcal{R}$ be an invertible representation with $\Psi$ being its inverse. Let $p_\Phi^{T=1}(\mathbf{r})$, $p_\Phi^{T=0}(\mathbf{r})$ be as defined before. Let $h : \mathcal{R} \times \{0,1\} \to \mathcal{Y}$, $u := Pr(T = 1)$ and $\mathcal{H}$ be the family of binary functions.*

*Let loss function $L$ be the squared loss such that $L(y_1, y_2) = (y_1 - y_2)^2$. Then we have:*

$$\epsilon_{PEHE}(h, \Phi)$$
$$\leq 2(\epsilon_{CF}(h, \Phi) + \epsilon_F(h, \Phi) - 2\sigma_y^2) \tag{40}$$
$$\leq 2(\epsilon_F^{T=1}(h, \Phi) + \epsilon_F^{T=0}(h, \Phi) + \frac{K}{2}d_\mathcal{H}(p_\Phi^{T=1}, p_\Phi^{T=0}) - 2\sigma_y^2) \tag{41}$$

*Let loss function $L$ be the absolute loss such that $L(y_1, y_2) = |y_1 - y_2|$. Then we have:*

$$\epsilon_{PEHE}(h, \Phi)$$
$$\leq \epsilon_{CF}(h, \Phi) + \epsilon_F(h, \Phi) - 2A_y \tag{42}$$
$$\leq \epsilon_F^{T=1}(h, \Phi) + \epsilon_F^{T=0}(h, \Phi) + \frac{K}{2}d_\mathcal{H}(p_\Phi^{T=1}, p_\Phi^{T=0}) - 2A_y \tag{43}$$

**Proof** *Inequality (40) is by Lemma 5, inequality (41) is by Lemma 1 and Lemma 4; Inequality (42) is by Lemma 6, inequality (43) is by Lemma 1 and Lemma 4;* $\qquad\square$

Obviously, when using Wasserstein distance, there are various versions of bounds for different loss functions as long as they satisfy the (relaxed) triangle inequality and assumptions about $\ell_{h,\Phi}$ in Theorem 1. Similarly, when using $\mathcal{H}$-divergence, there are also various versions of bounds for loss functions that satisfy Assumption 1 and the (relaxed) triangle inequality.

For an empirical sample and a family of representations and hypotheses, we can further upper bound $\epsilon_F^{T=0}$ and $\epsilon_F^{T=1}$ by their respective empirical losses and a model complexity term using standard arguments (Shalev-Shwartz & Ben-David, 2014). Both the Wasserstein distance and $\mathcal{H}$-divergence can be consistently estimated from finite samples (Sriperumbudur et al., 2012; Ben-David et al., 2006; 2010).

## A.4 ILLUSTRATIVE EXAMPLES

**Examples for the motivation for decomposed patterns.** To explain the dilemma between representation balancing and outcome prediction, we give two intuitive examples below to help readers better understand the motivation and importance of involving decomposed patterns in representation balancing models.

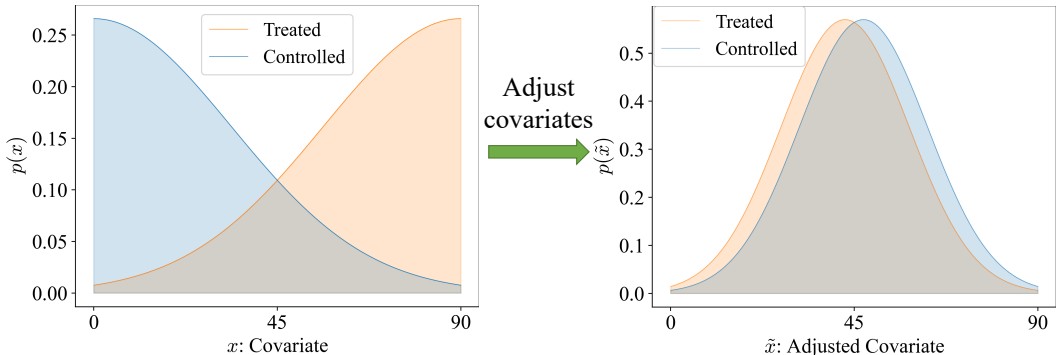

Figure 5: Example for illustrating the importance of decomposed patterns.

**Example 1.**    Suppose there is a vaccine to prevent some kind of disease. Let $X$ denote the covariate (age), $T = 1$ denote the treatment (getting vaccinated), $T = 0$ denote the control (not getting vaccinated), and $Y$ denote the outcome (probability of getting the disease). Suppose that the vaccine is assigned according to age, and we have found that the older, the higher the probability of getting the disease. The left graph in Figure 5 shows the distribution of pre-balancing covariate $X$ for treated and controlled groups, which indicates that vaccines are more likely to distribute to older people. Technically, the pre-balancing data preserve the outcome-predictive information: if we want to estimate $Y$ using the covariate $X$, we are confident that people in the treatment/control group (orange/blue) are susceptible/unsusceptible to the disease since they are older/younger. The right panel of Figure 5 shows the distribution of the adjusted covariate $\tilde{X}$, over which the distributions of treated and controlled groups are highly balanced. In this case, however, the distribution of $\tilde{X}$ is too balanced, making it hard to distinguish the treatment samples from the control samples. Consequently, if we want to estimate $Y$ using $\tilde{X}$, we may get confused about which group is susceptible to the disease because the distributions of $\tilde{X}$ are almost identical between the treated and controlled groups. Therefore, only considering balancing patterns can result in a loss of outcome-predictive information.

**Example 2.**    Following the example above, allow us to give a special but more intuitive instance. Imagine two men are entirely identical other than age, of whom one is older (treatment, T) and the other is younger (control, C). So we can use the covariate age to distinguish between T and C. We also found that the older one is susceptible to the disease. However, once their ages are mapped to some representations such that their representations are over-balanced, even identical, then such representations can be useless to distinguish who is T and who is C. As a result, it will be difficult for such representations to be used for estimating who is susceptible to the disease. Therefore, over-balanced representations may lose outcome-predictive information.

In summary, on the one hand, involving representation balancing can benefit treatment effect estimation. On the other hand, if $p_\Phi^{T=1}$ and $p_\Phi^{T=0}$ are too balanced, a model may fail to preserve pre-balancing information that is useful to outcome predictions. Such a dilemma motivates us to incorporate PPBR and PDIG such that PPBR improves outcome prediction without harming representation balancing, and PDIG helps a model to achieve more balanced representations without harming outcome prediction.

### A.5    ADDITIONAL EXPERIMENTAL DETAILS

**Hyperparameters.**    In simulation studies, we ensure a fair comparison by fixing all the hyperparameters in all datasets across different models. The relevant details are stated in Table 4. In IHDP studies, to compare with the baseline model CFR-Wass (GNet), we remain the hyperparameters of INet, DGNet, DINet and the early stopping rule the same as those used in CFR-Wass Shalit et al. (2017). Since DIGNet is more complex than other four models, we adjust the hyperparameters of $\Phi_E$, $\Phi_G$, $\Phi_I$, $\alpha_1$, and $\alpha_2$ for DIGNet as Shalit et al. (2017) do. The relevant details are stated in Table 5.

Table 4: Hyperparameters of different models in simulation studies.

| | $\Phi_E$ | $\Phi_G$ | $\Phi_I$ | $\pi$ | $h^1$ | $h^0$ | $\alpha_1$ | $\alpha_2$ | batchsize | iteration | learning rate | learning rate for $\pi$ |
|---|---|---|---|---|---|---|---|---|---|---|---|---|
| Gnet | $(100,100,100,100)$ | – | – | – | $(100,100)$ | $(100,100)$ | $0.1$ | – | $100$ | $300$ | $1e^{-3}$ | – |
| Inet | $(100,100,100,100)$ | – | – | $(100,100,100)$ | $(100,100)$ | $(100,100)$ | – | $0.1$ | $100$ | $300$ | $1e^{-3}$ | $1e^{-4}$ |
| DGNet | $(100,100,100,100)$ | $(100,100)$ | – | – | $(100,100)$ | $(100,100)$ | $0.1$ | – | $100$ | $300$ | $1e^{-3}$ | – |
| DINet | $(100,100,100,100)$ | – | $(100,100)$ | $(100,100,100)$ | $(100,100)$ | $(100,100)$ | – | $0.1$ | $100$ | $300$ | $1e^{-3}$ | $1e^{-4}$ |
| DIGNet | $(100,100,100,100)$ | $(100,100)$ | $(100,100)$ | $(100,100,100)$ | $(100,100)$ | $(100,100)$ | $0.1$ | $0.1$ | $100$ | $300$ | $1e^{-3}$ | $1e^{-4}$ |

Table 5: Hyperparameters of different models in IHDP experiments.

| | $\Phi_E$ | $\Phi_G$ | $\Phi_I$ | $\pi$ | $h^1$ | $h^0$ | $\alpha_1$ | $\alpha_2$ | batchsize | iteration | learning rate | learning rate for $\pi$ |
|---|---|---|---|---|---|---|---|---|---|---|---|---|
| Gnet | $(100,100,100,100)$ | – | – | – | $(100,100,100)$ | $(100,100,100)$ | $1$ | – | $100$ | $600$ | $1e^{-3}$ | – |
| Inet | $(100,100,100,100)$ | – | – | $(200,200,200)$ | $(100,100,100)$ | $(100,100,100)$ | – | $1$ | $100$ | $600$ | $1e^{-3}$ | $1e^{-3}$ |
| DGNet | $(100,100,100,100)$ | $(100,100)$ | – | – | $(100,100,100)$ | $(100,100,100)$ | $1$ | – | $100$ | $600$ | $1e^{-3}$ | – |
| DINet | $(100,100,100,100)$ | – | $(100,100)$ | $(200,200,200)$ | $(100,100,100)$ | $(100,100,100)$ | – | $1$ | $100$ | $600$ | $1e^{-3}$ | $1e^{-3}$ |
| DIGNet | $(100,100,100,100,100,100)$ | $(100,100,100)$ | $(100,100,100)$ | $(200,200,200)$ | $(100,100,100)$ | $(100,100,100)$ | $0.1$ | $1$ | $100$ | $600$ | $1e^{-3}$ | $1e^{-3}$ |

**Device.** All the experiments are run on Dell 7920 with 1x 16-core Intel Xeon Gold 6250 3.90GHz CPU and 3x NVIDIA Quadro RTX 6000 GPU.

Table 6: Additional comparisons on IHDP dataset.

| | Training set | | Test set | |
|---|---|---|---|---|
| | $\sqrt{\epsilon_{PEHE}}$ | $\epsilon_{ATE}$ | $\sqrt{\epsilon_{PEHE}}$ | $\epsilon_{ATE}$ |
| OLS/LR$_1$ (Johansson et al., 2016) | $5.8 \pm .3$ | $.73 \pm .04$ | $5.8 \pm .3$ | $.94 \pm .06$ |
| OLS/LR$_2$ (Johansson et al., 2016) | $2.4 \pm .1$ | $.14 \pm .01$ | $2.5 \pm .1$ | $.31 \pm .02$ |
| k-NN (Crump et al., 2008) | $2.1 \pm .1$ | $.14 \pm .01$ | $4.1 \pm .2$ | $.79 \pm .05$ |
| BART (Chipman et al., 2010) | $2.1 \pm .1$ | $.23 \pm .01$ | $2.3 \pm .1$ | $.34 \pm .02$ |
| CF (Wager & Athey, 2018) | $3.8 \pm .2$ | $.18 \pm .01$ | $3.8 \pm .2$ | $.40 \pm .03$ |
| CEVAE (Louizos et al., 2017) | $2.7 \pm .1$ | $.34 \pm .01$ | $2.6 \pm .1$ | $.46 \pm .02$ |
| SITE (Yao et al., 2018) | $.69 \pm .0$ | $.22 \pm .01$ | $.75 \pm .0$ | $.24 \pm .01$ |
| GANITE (Yoon et al., 2018) | $1.9 \pm .4$ | $.43 \pm .05$ | $2.4 \pm .4$ | $.49 \pm .05$ |
| BLR (Johansson et al., 2016) | $5.8 \pm .3$ | $.72 \pm .04$ | $5.8 \pm .3$ | $.93 \pm .05$ |
| BNN (Johansson et al., 2016) | $2.2 \pm .1$ | $.37 \pm .03$ | $2.1 \pm .1$ | $.42 \pm .03$ |
| TARNet (Shalit et al., 2017) | $.88 \pm .0$ | $.26 \pm .01$ | $.95 \pm .0$ | $.28 \pm .01$ |
| CFR-Wass (GNet) (Shalit et al., 2017) | $.73 \pm .0$ | $.12 \pm .01$ | $.81 \pm .0$ | $.15 \pm .01$ |
| Dragonnet (Shi et al., 2019) | $1.3 \pm .4$ | $.14 \pm .01$ | $1.3 \pm .5$ | $.20 \pm .05$ |
| DKLITE (Zhang et al., 2020) | $.52 \pm .0$ | – | $.65 \pm .03$ | – |
| CFR-ISW (Hassanpour & Greiner, 2019a) | – | – | $.70 \pm .0$ | $.19 \pm .03$ |
| BWCFR-OW (Assaad et al., 2021) | – | – | $.65 \pm .0$ | $.18 \pm .01$ |
| BWCFR-MW (Assaad et al., 2021) | – | – | $.63 \pm .0$ | $.19 \pm .01$ |
| BWCFR-TruncIPW (Assaad et al., 2021) | – | – | $.63 \pm .0$ | $.19 \pm .01$ |
| MBRL (Huang et al., 2022a) | $.52 \pm .0$ | $.12 \pm .01$ | $.57 \pm .0$ | $.13 \pm .01$ |
| DIGNet (Ours) | $\mathbf{.42 \pm .0}$ | $\mathbf{.11 \pm .01}$ | $\mathbf{.45 \pm .0}$ | $\mathbf{.12 \pm .01}$ |

**Additional IHDP results.** The selection bias of different simulated datasets is illustrated in Figure 6. We compare DIGNet with more baselines in Table 6. Note that − indicates either the result is not reproducible or the original paper does not report relevant values. We collect baseline methods that focus on treatment effect estimation, especially methods using deep representation learning techniques, from recent machine learning conferences (e.g., ICML, NeurIPS, ICLR, AISTATS, and PRICAI)

**Analysis for training time and training stability.** We record the time it took for different models to run through 100 IHDP datasets, and each model is trained within 600 epochs. Following Shalit et al. (2017), all models adopt the early stopping rule. We also record the average early stopping epoch on 100 runs and the actual time on 100 runs, where (actual time) = (total time) × (average early stopping epoch)/600. Not surprisingly, GNet took the least amount of time with 3096 seconds since the objective of GNet is the simplest. However, it is very interesting that the proposed methods, DGNet and DINet, are the first two to early stop. As a result, though DGNet and DINet have multi-objectives, they spent less actual training time but achieved better ITE estimation compared to GNet and INet. Since GNet and INet are actually DGNet and DINet with PPBR ablated, we find that PPBR component can help a model achieve better ITE estimates with less time. In addition, we find that DIGNet spent the longest time to optimize since it has the most complex objective. To further study the stability of the model training, we also plot the metrics $\sqrt{\epsilon_F}$, Wass, $\hat{d}_{\mathcal{H}}$, and $\sqrt{\epsilon_{PEHE}}$ for the first 100 epochs of each model on the first IHDP dataset. We find that the training process

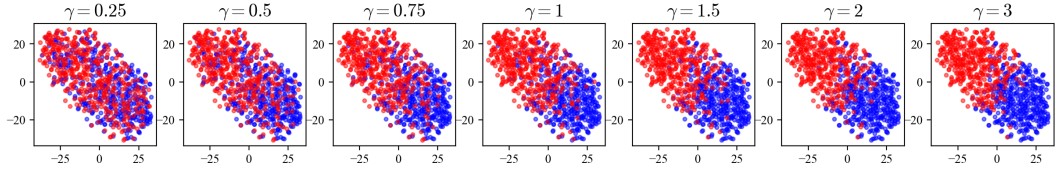

Figure 6: T-SNE visualizations of the covariates as $\gamma$ varies. Red represents the treatment group and blue represents the control group. A larger $\gamma$ indicates a greater imbalance between the two groups.

of DIGNet is stable, even steadier than GNet and INet. From this perspective, we haven't seen a difficulty of optimizing DIGNet.

Table 7: Training time records on 100 IHDP datasets.

| Model | Time for 600 epochs | Avg early stopping | Actual time | $\sqrt{\epsilon_{PEHE}}$ on test set |
|---|---|---|---|---|
| GNet | 3096s | 240.61 | 1241s | 0.77±0.18 |
| INet | 4042s | 254.19 | 1712 | 0.72±0.11 |
| DGNet | 3775s | 169.17 | 1064s | 0.60±0.09 |
| DINet | 3212s | 157.98 | 846s | 0.60±0.11 |
| DIGNet | 4984s | 226.76 | 1884s | 0.45±0.04 |

Figure 7: Training loss plots for the first 100 epochs on the first IHDP dataset.

## A.6 OBJECTIVES FOR DIFFERENT MODELS

**Objective of GNet.**

$$\min_{\Phi_E, h^t} \quad \mathcal{L}_y(\mathbf{x}, \mathbf{t}, \mathbf{y}; \Phi_E, h^t) + \alpha_1 \mathcal{L}_G(\mathbf{x}, \mathbf{t}; \Phi_E).$$

**Objective of INet.**

$$\max_{\pi} \quad \alpha_2 \mathcal{L}_I(\mathbf{x}, \mathbf{t}; \Phi_E, \pi),$$

$$\min_{\Phi_E, h^t} \quad \mathcal{L}_y(\mathbf{x}, \mathbf{t}, \mathbf{y}; \Phi_E, h^t) + \alpha_2 \mathcal{L}_I(\mathbf{x}, \mathbf{t}; \Phi_E, \pi).$$

**Objective of DINet.** Note that similar to DIGNet, the pre-balancing patterns are preserved by only updating $\Phi_I$ but fixing $\Phi_E$ in the second step.

$$\max_{\pi} \quad \alpha_2 \mathcal{L}_I(\mathbf{x}, \mathbf{t}; \Phi_I \circ \Phi_E, \pi),$$

$$\min_{\Phi_I} \quad \alpha_2 \mathcal{L}_I(\mathbf{x}, \mathbf{t}; \Phi_I \circ \Phi_E, \pi),$$

$$\min_{\Phi_E, \Phi_I, h^t} \quad \mathcal{L}_y(\mathbf{x}, \mathbf{t}, \mathbf{y}; \Phi_E \oplus (\Phi_I \circ \Phi_E), h^t).$$

**Objective of DGNet.** Note that similar to DIGNet, the pre-balancing patterns are preserved by only updating $\Phi_G$ but fixing $\Phi_E$ in the first step.

$$\min_{\Phi_G} \quad \alpha_1 \mathcal{L}_G(\mathbf{x}, \mathbf{t}; \Phi_G \circ \Phi_E),$$

$$\min_{\Phi_E, \Phi_G, h^t} \quad \mathcal{L}_y(\mathbf{x}, \mathbf{t}, \mathbf{y}; \Phi_E \oplus (\Phi_G \circ \Phi_E), h^t).$$

**Objective of DIGNet.**

$$\min_{\Phi_G} \quad \alpha_1 \mathcal{L}_G(\mathbf{x}, \mathbf{t}; \Phi_G \circ \Phi_E),$$

$$\max_{\pi} \quad \alpha_2 \mathcal{L}_I(\mathbf{x}, \mathbf{t}; \Phi_I \circ \Phi_E, \pi),$$

$$\min_{\Phi_I} \quad \alpha_2 \mathcal{L}_I(\mathbf{x}, \mathbf{t}; \Phi_I \circ \Phi_E, \pi),$$

$$\min_{\Phi_E, \Phi_I, \Phi_G, h^t} \quad \mathcal{L}_y(\mathbf{x}, \mathbf{t}, \mathbf{y}; \Phi_E \oplus (\Phi_I \circ \Phi_E) \oplus (\Phi_G \circ \Phi_E), h^t).$$

