# OpenReview forum: "Representation Balancing with Decomposed Patterns for Treatment Effect Estimation"
_ICLR.cc/2023/Conference — Submitted to ICLR 2023_

### Official Review · Reviewer_eA2v · 2022-10-23

**Confidence:** 2
**Correctness:** 3
**Technical Novelty And Significance:** 2
**Empirical Novelty And Significance:** 2
**Recommendation:** 6

**Clarity, Quality, Novelty And Reproducibility:**

The paper is generally well-written, and the contributions are clearly stated. The proposed method seems to be a useful treatment effect approach. The simulation results look promising.

**Strength And Weaknesses:**

Strength:

The proposed method is supported by two bounds of the expected estimation error of heterogeneous treatment effects. The first bound of treatment effect estimation error uses the Wasserstein distance between treated and control groups. The second bound of treatment effect estimation error uses H-divergence as a measure of propensity confusion.

Weakness:

The connection between the two components (PDIG and PPBR) could have been better elaborated. For example, how is PDIG helpful for PPBR? Furthermore, it would be nice to provide an example of why it is important to account for pre-balancing representations for outcome prediction. A simple numerical example could be useful to understand intuition.


**Summary Of The Paper:**

This paper proposes a framework for treatment effect estimation that consists of two components. The first is to balance representations by minimizing group distance and maximizing individual propensity confusion (i.e., PDIG). The second is to use both pre-balancing and post-balancing representations for outcome prediction (i.e., PPBR).

**Summary Of The Review:**

This paper proposes a method for treatment effect estimation. This method balances representations, and uses both pre-balancing and post-balancing representations for outcome prediction. This method is demonstrated to have good performance in simulations.

---

### Official Review · Reviewer_ncJx · 2022-10-24

**Confidence:** 4
**Clarity, Quality, Novelty And Reproducibility:** See above.
**Correctness:** 3
**Technical Novelty And Significance:** 2
**Empirical Novelty And Significance:** 2
**Recommendation:** 6

**Strength And Weaknesses:**

**Strengths**
- The paper is well-written and easy to follow
- The proposed approach seems easy to implement
- Leveraging both propensity score and balanced representation in a unified framework is interesting

**Weakness**

*Novelty*
- The paper seems to be a straightforward combination of DANN (Ganin et al., 2016) and CFR (Shalit et al., 2017)
- The paper should highlight how the proposed approach differs from related works

*Unconvincing Experiments*
- It may be challenging to optimize the four objectives, which also include adversarial learning
- The paper should also compare with similar approaches incorporating balanced representation and propensity score learning, *e.g.*, Zhang et al., 2020; Assaad et al., 2021; Huang et al., 2022a;  Hassanpour & Greiner, 2019b; *etc.*
- How are the $\alpha$ weights selected?
- The paper should provide an expanded discussion  on the chosen baselines and Table 3 results

**Summary Of The Paper:**

The paper proposes a framework that accounts for observed confounding in treatment effect estimation via (i) balanced representation learning of treatment groups and (ii) adversarial propensity score learning. Experimental results on synthetic and semi-synthetic data sets demonstrate improved treatment effect estimation.

**Summary Of The Review:**

The paper combines previously proposed DANN and  CFR in a unified framework. However, the performance gain of the proposed approach seems marginal when compared to the  DANN framework alone (DINet). Additionally, several baselines are missing from the experimental evaluation.

**Post Author (s) Response**
- The author (s) addressed most of my concerns, and I have adjusted my score accordingly.

---

### Official Review · Reviewer_JQVc · 2022-10-28

**Confidence:** 3
**Correctness:** 4
**Technical Novelty And Significance:** 3
**Empirical Novelty And Significance:** 2
**Recommendation:** 6

**Clarity, Quality, Novelty And Reproducibility:**

I found the paper to be well written and organized. As I mention above this paper follows prior work fairly directly, but improves upon prior art so I think it is still a valuable contribution.

**Strength And Weaknesses:**

The strengths of this paper are in the relative simplicity and efficacy of the proposed approach. The use of H-divergences is both natural and well motivated with theory. I thought the authors did a nice job of presenting both the motivation and solution as well. I think the empirical results are also compelling.

In terms of weakness, the main issue for me is a relative lack of novelty, though I don't think that is enough to vote for rejection at this time.

**Summary Of The Paper:**

This work seeks to improve upon representation learning modeling for inferring individualized treatment effects which employs a discrepancy constraint between the representations of the treatment and control groups. The authors propose to use H-divergences to achieve this. This is motivated by a bound on the ITE error by an H-divergence. Empirical results show gains relative to prior art.

**Summary Of The Review:**

As I mention above, I think this is a nice, simple idea with good execution. The authors do a nice job of explaining the relative merits of using an H-divergence and provide compelling empirical evidence for their claims as well. While this work may border on incremental in terms of novelty, I think it is still a valuable addition to the literature.

---

### Decision · Program_Chairs · 2023-01-20

**Decision:**

Reject

**Justification For Why Not Higher Score:**

The method and results are incremental.

**Justification For Why Not Lower Score:**

N/A

**Metareview: Summary, Strengths And Weaknesses:**

Since this was a borderline paper, I have carefully read the paper myself. I agree with some of the major concerns of the reviewers, even after reading the author responses.

I do think that the contribution is incremental in terms of novelty. The authors essentially introduce an extra divergence distance measure on top of the wasserstein distance, in the same spirit as DCGans, with a discrimination loss, that amounts to a propensity estimator.

The theoretical result seems to be technically straightforward and follow similar (of course not identical) analysis to the proof approach in Shalit et al, plus extra basic properties of the H-divergence.

The final method proposed is an ad-hoc combination of the two distance measures and the experimental performance is only marginally better than the prior best method (the MBRL method). The real world evaluation is only done on IHDP which is a very over-analyzed data set and hence not convincing that the same marginally better performance would stand other data sets qualitatively different in nature. For instance, I would advice the authors to try more extensive ITE benchmarks, such as the semi-synthetic datasets offered by the RealCause package.